# Sex-biased genetic programs in liver metabolism and liver fibrosis are controlled by EZH1 and EZH2

**Dana Lau-Corona** [1], **Woo Kyun Bae** [2,3], **Lothar Hennighausen** [2], **David J. Waxman** [1]*

**1** Department of Biology and Bioinformatics Program, Boston University, Boston, Massachusetts, United States of America, **2** Laboratory of Genetics and Physiology, National Institute of Diabetes and Digestive and Kidney Diseases, National Institutes of Health, Bethesda, Maryland, United States of America, **3** Department of Internal Medicine, Chonnam National University Medical School, Gwangju, Korea

* djw@bu.edu

**Data Availability Statement:** All raw and processed RNA-seq and ChIP-seq data for 7-week floxed control and E1/E2-KO mice are available under accession number GSE110934 (https://www.ncbi.nlm.nih.gov/geo/query/acc.cgi?acc=

## Abstract

Sex differences in the incidence and progression of many liver diseases, including liver fibrosis and hepatocellular carcinoma, are associated with sex-biased hepatic expression of hundreds of genes. This sexual dimorphism is largely determined by the sex-specific pattern of pituitary growth hormone secretion, which controls a transcriptional regulatory network operative in the context of sex-biased and growth hormone-regulated chromatin states. Histone H3K27-trimethylation yields a major sex-biased repressive chromatin mark deposited at many strongly female-biased genes in male mouse liver, but not at male-biased genes in female liver, and is catalyzed by polycomb repressive complex-2 through its homologous catalytic subunits, Ezh1 and Ezh2. Here, we used *Ezh1*-knockout mice with a hepatocyte-specific knockout of *Ezh2* to investigate the sex bias of liver H3K27-trimethylation and its functional role in regulating sex-differences in the liver. Combined hepatic Ezh1/Ezh2 deficiency led to a significant loss of sex-biased gene expression, particularly in male liver, where many female-biased genes were increased in expression while male-biased genes showed decreased expression. The associated loss of H3K27me3 marks, and increases in the active enhancer marks H3K27ac and H3K4me1, were also more pronounced in male liver. Further, Ezh1/Ezh2 deficiency in male liver, and to a lesser extent in female liver, led to up regulation of many genes linked to liver fibrosis and liver cancer, which may contribute to the observed liver pathologies and the increased sensitivity of these mice to hepatotoxin exposure. Thus, Ezh1/Ezh2-catalyzed H3K27-trimethyation regulates sex-dependent genetic programs in liver metabolism and liver fibrosis through its sex-dependent effects on the epigenome, and may thereby determine the sex-bias in liver disease susceptibility.

## Author summary

Sex-differences in the expression of genes in liver have a direct impact on liver diseases whose incidence and severity is sex-biased, and is controlled by hormones that regulate chemical alterations to histone proteins used to package chromosomal DNA. However, a

GSE110934). RNA-seq data for carbon tetrachloride-treated control and E1/E2-KO male mouse livers, and for 8-month control and E1/E2-KO male mouse livers are available under accession number GSE53627 (https://www.ncbi.nlm.nih.gov/geo/query/acc.cgi?acc=GSE53627).

**Funding:** Supported in part by grants DK121998 and DK33765 from the NIH (to DJW) and by the NIH Intramural Research Program (to LH). The funding sponsor played no role in the design of the study and collection, analysis, and interpretation of data or in writing the manuscript.

**Competing interests:** The authors declare that they have no competing interests.

direct demonstration of the functional importance of such sex differences in histone protein modifications has been elusive. Here, we address this question using a mouse model deficient in two enzymes, Ezh1/Ezh2, which generate the histone repressive mark H3K27me3. Remarkably, although H3K27me3 marks are formed by Ezh1/Ezh2 throughout the genome, loss of liver Ezh1/Ezh2 preferentially disrupts the control of sex-biased genes, with expression increasing in male mouse liver for many female-biased genes and decreasing for many male-biased genes. Sex-biased H3K27me3 repressive marks were abolished, and there was a gain of active histone marks at gene enhancers. We also found increased expression of many genes associated with liver fibrosis and hepatocellular carcinoma, which may help explain the increased sensitivity of Ezh1/Ezh2-deficient livers to hepatotoxic chemicals whose exposure may lead to sex differences in liver disease incidence and susceptibility. Thus, our findings highlight the potential role of sex differences in histone modifications catalyzed by Ezh1/Ezh2 in widespread sex differences in liver diseases.

## Introduction

Liver disease shows marked sex differences. Hepatocellular carcinoma incidence and mortality is three times higher in men than in women [1, 2], and male mice are more susceptible to chemical-induced hepatic carcinogenesis [3]. Males are also more susceptible to non-alcoholic fatty liver disease, non-alcoholic steatohepatitis, and liver fibrosis than females [4–8]. Underlying these sex-biased (sex-enriched) phenotypical differences are hundreds of genes that are expressed in liver in a sexually dimorphic manner. This dimorphism is largely regulated by the sex-specific patterns of pituitary secretion of growth hormone (GH), which is intermittent (pulsatile) in males and persistent (near continuous) in females [9, 10]. The sex-specific effects of GH require the GH-activated transcription factor STAT5b [11, 12]. STAT5b regulates sex-specific liver gene expression in collaboration with several sex-biased, GH-responsive transcription factors [13, 14], which operate in the context of sex-biased long non-coding RNAs [15], microRNAs [16], and sex-biased chromatin states [17–19]. Trimethylation of histone H3 at lysine 27 (H3K27me3) has been identified as a sex-biased chromatin mark in mouse liver: a striking male bias in the density of H3K27me3 marks is seen across many highly female-biased genes in male liver, but there is no corresponding sex bias at male-biased genes in female liver [17]. Continuous infusion of male mice with GH, which overrides endogenous male plasma GH pulses and imposes a female-like hormonal environment, leads to a loss of hepatic H3K27me3 marks at highly female-biased genes in association with induction of female-biased gene expression [19]. However, a direct demonstration of the functional importance of such sex differences in H3K27me3 marks has been elusive.

 H3K27me3 is a hallmark of transcriptional silencing, and is deposited by polycomb repressive complex-2 (PRC2), a protein complex involved in cell differentiation, cell-specific identity and cell proliferation [20, 21]. PRC2 has three core components, Suz12, Eed, and either Ezh1 or Ezh2. Ezh1 and Ezh2 both contain a SET domain, which is required for methylation of histone H3 lysine-27, and can each serve as the catalytic subunit of PRC2. PRC2 facilitates transcriptional repression by recruiting protein complexes that recognize H3K27me3 and induce chromatin compaction [22]. Ezh1 and Ezh2 have complementary and compensatory functions and share an overlapping set of target genes [23, 24]. Ezh1 has lower methyltransferase activity than Ezh2, and its expression persists in adult tissues, in contrast to Ezh2, which is preferentially expressed in embryonic and highly proliferative tissues [23]. Deletion of the SET domain of Ezh2 in embryonic mouse liver results in a significant reduction of liver size and hepatic progenitor cell expansion, impairing liver differentiation and maturation [25].

PRC2 represses the expression of tumor suppressor genes through both H3K27me3-dependent and H3K27me3-independent mechanisms, promoting tumor formation [26, 27]. Increased levels of Ezh2 and H3K27me3 are found in hepatocellular carcinoma (HCC) and are associated with metastasis and poor prognosis [28]. Ezh2 silences several tumor suppressor miRNAs that are down-regulated in liver cancer [27] and it interacts with highly expressed oncogenic long non-coding RNAs (lncRNAs) to repress target genes in HCC [29, 30]. However, Ezh1 and Ezh2 can also exert anti-tumor effects [31, 32]. Notably, the beneficial effects of Ezh1 and Ezh2 are apparent in adult mouse liver, where the functional loss of both genes induces gene dysregulation accompanied by a severe decrease in liver function, impaired liver regeneration, and induction of liver fibrosis [33], which often leads to development of HCC [34]. Liver steatosis, fibrosis and HCC development are also induced following disruption of GH-STAT5 signaling in mouse liver [35–37], where the metabolic effects of GH signaling loss linked to fatty liver development are more pronounced in males than females [38]. Together, these findings raise the possibility that the sex-dependent pathologies seen in GH signaling-disrupted liver, could, in part, involve the loss of GH-regulated and Ezh1/Ezh2-dependent deposition of H3K27me3 marks required for physiologically balanced expression of sex-biased genes in the liver.

Here, we use an *Ezh1*-knockout mouse model with a hepatocyte-specific knockout of *Ezh2* to investigate the role of H3K27me3 in regulating sex-biased gene expression in mouse liver, and the potential impact of this regulation on sex-biased susceptibility to liver disease. Our findings reveal a significant sex-bias in the impact of Ezh1/Ezh2 loss, with a striking preference for depletion of H3K27me3 marks and increased expression of female-biased genes in Ezh1/Ezh2-deficient male liver. Hepatic Ezh1/Ezh2 deficiency is also shown to down regulate many male-biased genes, presumed as a secondary response to the disruption of female-biased gene expression. Finally, we show that many genes associated with liver fibrosis and liver carcinogenesis are differentially responsive to the loss of Ezh1/Ezh2 in male compared to female liver, which may contribute to the observed sex-differences in the incidence and progression of liver cancer.

## Results

### Expression of Ezh1 and Ezh2 in postnatal mouse liver

Ezh1 was unchanged in expression in postnatal male and female mouse liver between 2 and 8 weeks of age. In contrast, *Ezh2* expression peaked at 2 weeks, when hepatocytes are still proliferating [39], and then progressively declined with age in both sexes (Fig 1A). This is consistent with the preferential expression of *Ezh2* in highly proliferative cells and its decline in postnatal development seen in other mouse tissues [23]. A moderate (~2-fold) female bias in expression of *Ezh1*, but not *Ezh2*, was seen at 2 weeks of age. *Ezh1* but not *Ezh2* mRNA levels were greatly reduced in livers of both sexes in 7-week-old *Ezh1*-knockout mice with hepatocyte-specific inactivation of *Ezh2* (abbreviated as 'E1/E2-KO' or 'DKO' (double-knockout) livers) (Fig 1B, Fig 1C). However, *Ezh2* was functionally knocked out, as transcription of the exons encoding the SET domain, which is essential for histone H3K27 methyltransferase activity [24], was undetectable (Fig 1B, *red box*). Western blotting verified the expression of Ezh2 protein in both male and female liver and its ablation upon hepatocyte-specific knockout (S1 Fig).

### Female-biased genes are preferentially de-repressed in Ezh1/Ezh2-deficient male liver

RNA-seq revealed that 1,355 (12%) of 11,491 liver-expressed genes are differentially expressed between E1/E2-KO and control mouse liver. A majority (72%) of the differentially expressed genes were up-regulated in the absence of Ezh1 and Ezh2, as is expected for a genetic

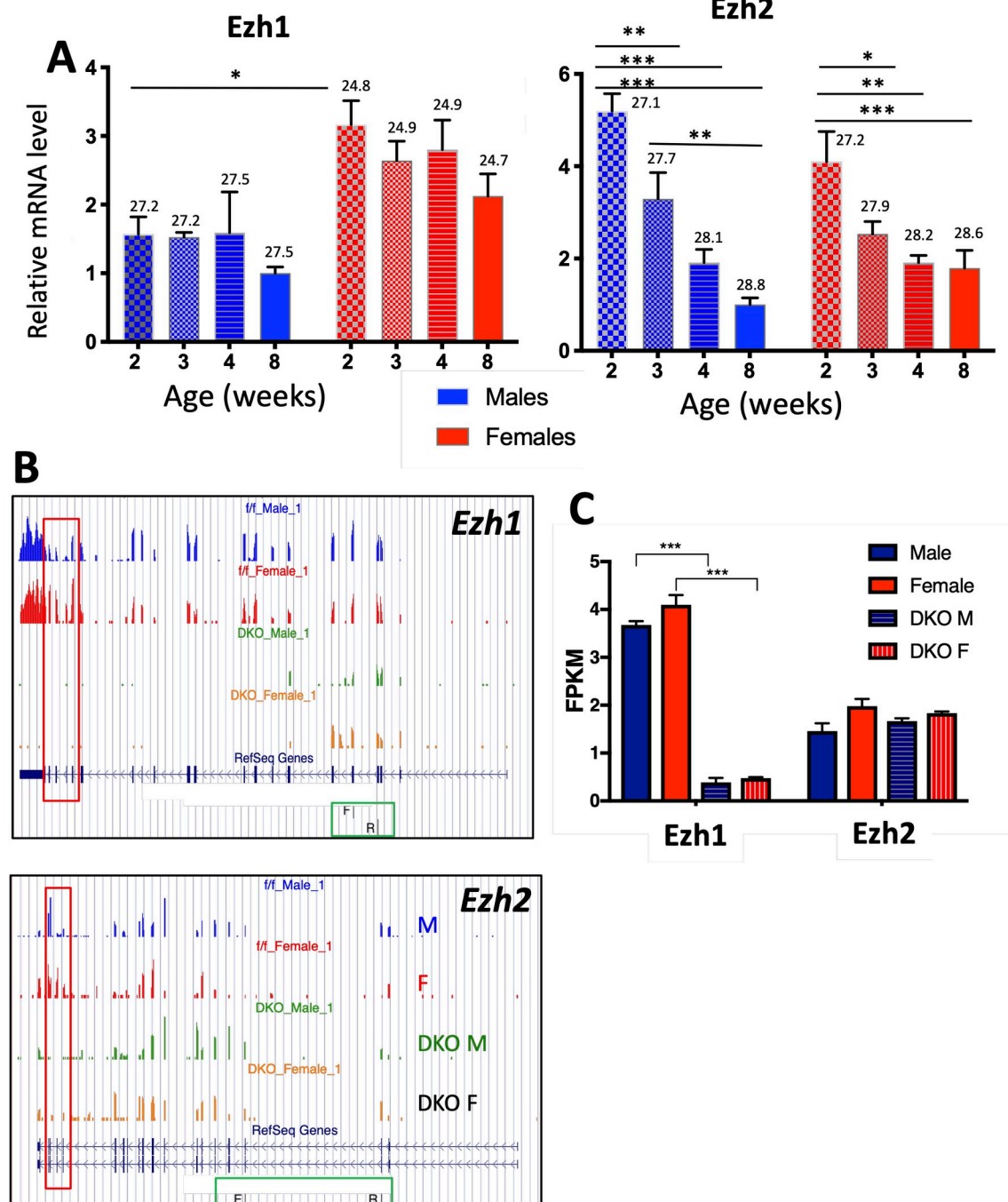

**Fig 1. *Ezh1* and *Ezh2* expression in wild-type pre-pubertal and young adult liver and in E1/E2-KO mouse liver.** (**A**) Relative expression levels of *Ezh1* and *Ezh2* determined by RT-qPCR in male and female mouse livers at 2, 3, 4 and 8 weeks of age. Data shown are mean ± SEM for n = 3 (*Ezh1*) or n = 6 (*Ezh2*) individual livers per group. Primers used are shown in S1A Table and their location marked with green boxes in Fig 1B. Significance values by 2-way ANOVA with Tukey's or Sidak's multiple comparisons test are indicated: * p < 0.05; ** p < 0.01 and *** p < 0.001. qPCR cycle threshold numbers (Ct values) are shown above each bar. (**B**) UCSC genome browser screenshots of RNA-seq BigWig tracks evidencing the absence in 7-week male and female E1/E2-KO livers of *Ezh1* and *Ezh2* sequence reads from several exons, including those that code for the SET domain (*Ezh1 exons 17–21 and Ezh2* exons 16–19; *red boxes*). (**C**) In the case of *Ezh1*, but not *Ezh2*, gene disruption leads to a significant decrease in overall normalized sequence reads, shown in units of FPKM. Data are mean ± SEM values for n = 3 individual livers per group. Significance values represent FDR values determined by EdgeR, *** p < 0.001. DKO, Ezh1/Ezh2 double knockout male (M) and female (F) mouse liver.

deficiency in the capacity for Ezh1/Ezh2-mediated deposition of H3K27me3 repressive chromatin marks. More genes showed dysregulated expression in E1/E2-KO males than in E1/E2-KO females (S3A Table). Moreover, both the up-regulated and the down-regulated gene sets were strongly enriched for sex-biased genes when compared to a background set of stringently-sex-independent genes that respond to E1/E2-KO in either sex (Fig 2A). DAVID analysis of the gene sets responsive to Ezh1/Ezh2 deficiency identified the following top terms and enrichment scores (ES) for the most significantly enriched gene clusters (S5 Table): for up-regulated genes: immunity (ES 6.7), cytochrome P450 (ES 6.2) and MHC class I (ES 4.3); and for down-regulated genes: endoplasmic reticulum (ES 9.6), oxidoreductase (ES 7.8), major urinary protein (ES 6.6), and retinol and drug metabolism (ES 5.4).

H3K27me3 is a major sex-biased repressive mark: it is found at many highly female-biased genes in male liver, but not at highly male-biased genes in female liver [17]. Supporting the functionality of the sex bias in H3K27me3 marks, many female-biased genes were up-regulated (de-repressed) in E1/E2-KO male liver, while few male-biased genes were up-regulated in E1/E2-KO female liver (Fig 2B). Genes up-regulated in E1/E2-KO male liver were strongly enriched for female-biased genes (ES = 10.1, p < 2.2E-16) compared to all liver-expressed genes, with 154 of the 250 up-regulated female-biased genes exclusively up-regulated in male E1/E2-KO liver, as compared to only 10 genes exclusively up-regulated in female E1/E2-KO liver (Fig 2C, *top left*). Male-biased genes were significantly enriched in the set of genes down-regulated in E1/E2-KO male liver (ES = 16.9, p < 2.2E-16), and there was a moderate enrichment of female-biased genes in the set of down-regulated in E1/E2-KO female liver (ES = 2.8, p = 5.7E-07) (Fig 2C; also see S3 Table). Thus, the loss of Ezh1/Ezh2 preferentially alters sex-biased gene expression in male liver, where many female-biased genes are induced (de-repressed) and male-biased genes are down-regulated. Stringently sex-independent genes did not show a significant sex bias in their response to the loss of liver Ezh1/Ezh2 (Fig 2C, *bottom*).

A majority of all E1/E2-KO-responsive female-biased genes lose sex-specificity in the absence of Ezh1/Ezh2 (191 of 294 genes (65%); Fig 2D), primarily due to their up regulation in male liver. Further, 64 of 146 (44%) E1/E2-KO-responsive male-biased genes lose sex-specificity, primarily due to their down regulation in male E1/E2-KO liver (Fig 2D; S2A Table, column U). The dysregulation of sex-biased genes in E1/E2-KO mouse liver can also be seen by comparing overall gene expression sex ratios in E1/E2-KO liver to control liver (S2 Fig). Whereas the loss of H3K27me3-based repression can directly explain the increased expression of female-biased genes in E1/E2-KO male liver, the decrease in male-biased gene expression is likely a secondary response to E1/E2-KO. This response could involve CUX2, a female-specific repressor of many male-biased genes [13, 40] that is induced 3.7-fold in male E1/E2-KO liver, insofar as 27% of the male-biased genes down-regulated in E1/E2-KO male mouse liver are direct targets of CUX2 (S2A Table, column AB).

Female-biased genes de-repressed in E1/E2-KO male liver include drug metabolizing enzymes genes, such as sulfotransferases (*Sult* genes) and cytochromes P450 (*Cyp* genes), e.g., the female-biased *Cyp2* family members *Cyp2b9*, *Cyp2c69*, (S3 Fig, S5 Fig). Unexpectedly, two female-biased *Cyp3* family genes were strongly induced in E1/E2-KO female but not E1/E2-KO male liver (*Cyp3a16*, 36-fold increase; *Cyp3a41a*, 4-fold increase) (S3 Fig, S6 Fig), thereby increasing the female-bias in their expression. Other highly female-biased genes, including *Sult2a1*, *Ntrk2*, *Ptgds*, *A1bg* and *Cyp3a44*, were strongly induced upon loss of Ezh1/Ezh2 in both male and female liver (S3–S5 Figs, S2A Table).

## Upstream regulators of Ezh1/Ezh2-dependent genes

Upstream Regulator Analysis identified STAT5b as the most significant upstream regulator of the set of female-biased genes de-repressed in E1/E2-KO male liver (p-value of overlap: 2.19E-

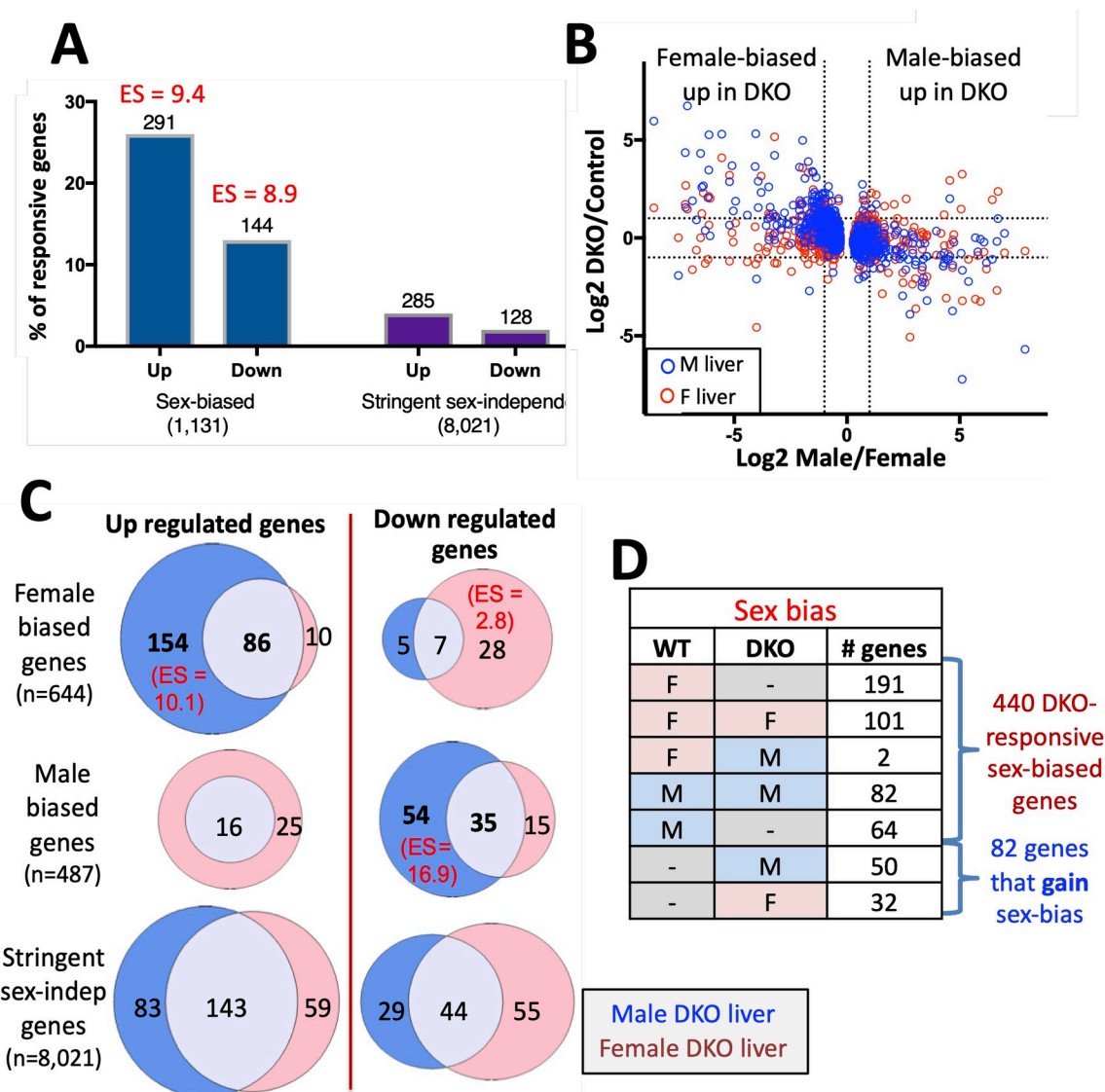

**Fig 2. E1/E2-KO de-represses female-biased genes in male liver.** (**A**) Percentage of all RefSeq genes that are liver-expressed (FPKM >1) and either sex-biased (male/female FDR < 0.01; n = 1,131 genes) (S2A Table) or stringently sex-independent (male/female |fold-change| < 1.2 and FDR > 0.1; n = 8,021 genes) (S2C Table) whose expression in E1/E2-KO liver is significantly changed compared to control liver (|fold-change| >1.5 at FDR <0.05) ('responsive genes'). The number of sex-biased genes that are up or down-regulated in E1/E2-KO liver and their enrichment (ES, enrichment score) compared to E1/E2-KO-responsive, stringently-sex independent genes is shown above each bar (p < 2.2E-16 for both enrichments). Overall, sex-biased genes comprised 38% of the genes dysregulated in E1/E2-KO liver (435 of 1,131 sex-biased genes; S3B Table) versus 5.1% of stringently-sex-independent genes (413 of 8,021 genes; S3C Table). (**B**) Log2 expression ratios for E1/E2-KO/control liver vs log2 sex-ratio for 1,131 sex-biased genes, determined in male (blue) and female (red) mouse liver. Dashed lines, significance cutoff values for each comparison. (**C**) Overlap of E1/E2-KO-responsive female-biased genes (and separately, male-biased genes) identified in male liver (blue shading) vs. female liver (pink shading). Overlaps are shown separately for genes that are up-regulated (*left*) and genes that are down-regulated (*right*) in E1/E2-KO liver. Shown at the bottom are the overlaps of stringent sex-independent genes that were E1/E2-KO responsive in male vs. female liver. Significant enrichment scores (ES) compared to a background set of all 11,491 liver-expressed genes (FPKM >1) are shown in parenthesis. Five genes responded to the loss of Ezh1/Ezh2 in the opposite direction in male vs. female liver and were excluded from these Venn diagrams: 4 female-biased genes up-regulated in male E1/E2-KO and down-regulated in female E1/E2-KO, and 1 male-biased gene up-regulated in female E1/E2-KO and down-regulated in male E1/E2-KO. DKO, Ezh1/Ezh2 double knockout mouse liver. (**D**) Number of E1/E2-KO-responsive genes that lose, maintain, or gain sex specificity in either male or female E1/E2-KO mouse liver. Two genes showed a reversal from female bias to male bias in E1/E2-KO liver. Sex bias is either: M, male; F, female; or not sex-biased (dashed line).

23), i.e., there is a highly significant enrichment for gene targets of STAT5b (S4A Table). Further, STAT5b showed a highly significant negative activation Z-score (Z = -4.6), indicating that up regulation of these genes is strongly associated with a loss of STAT5b function. Thus, STAT5b negatively regulates these female-biased genes in male liver, with Ezh1/Ezh2 deposition of H3K27me3 marks (see below) being a major mechanism for their repression by STAT5b. STAT5b was also identified as the strongest upstream regulator of the male-biased genes down-regulated in E1/E2-KO male liver (p-value of overlap: 9.76E-16) (S4B Table). A strong negative Z-score of -3.96 was found, in this case indicating that STAT5b positively regulates these male-biased genes in male liver, consistent with our prior findings [17, 18]. Other highly significant upstream regulators identified for female-biased genes up-regulated in male E1/E2-KO liver include key regulators of tumor necrosis factor and interferon signaling pathways, notably TNF, STAT1, IFNG, IFNB1, and IRF7 (p-values of overlap between 1.21E-17 and 1.58E-13), all with strong positive Z-scores (S4A Table), consistent with the enrichment of immunity pathways in the set of up-regulated genes by DAVID analysis (see above).

## Relationship between GH regulation and Ezh1/Ezh2 repression of female-biased genes in male liver

Hypophysectomy, which ablates GH and other circulating pituitary hormones, abolishes ~90% of liver sex differences. Furthermore, exogenous GH given either in pulses (male plasma GH pattern) or continuously (female-like GH pattern) substantially restores the corresponding sex-specific patterns of liver gene expression [41, 42], while GH given to intact male mice as a continuous infusion feminizes liver gene expression by inducing many female-biased genes and repressing male-biased genes. The induction of female-biased genes by continuous GH is associated with loss of H3K27me3 marks in male liver, as was shown for four highly female-biased genes [19]. To elucidate the relationship between H3K27me3-based repression of female-biased genes and regulation by the sex-specific patterns of GH secretion, we compared male liver gene responses to the loss of Ezh1/Ezh2, to the changes in gene expression seen following hypophysectomy [43] or after continuous GH infusion in mice for 14 days [19].

Loss of Ezh1/Ezh2 partially feminized the expression of a subset of strongly female-biased genes in male liver, as exemplified by the consistently strong, albeit incomplete up regulation of *Cyp2b9* in male E1/E2-KO liver (Fig 3A). No such up regulation was seen for another strongly female-biased gene, *Fmo3*. *Cyp2b9* and *Fmo3* represent two distinct classes of female-biased genes, which are defined based on their responses to hypophysectomy. Class I female-biased genes, such as *Fmo3*, require the female (near continuous) plasma GH pattern for full expression; consequently, loss of GH upon hypophysectomy represses those genes in female liver. In contrast, Class II female-biased genes, such as *Cyp2b9*, are repressed in male liver by the male pituitary hormone profile; consequently, hypophysectomy leads to their strong de-repression (i.e., induction) in male liver [41, 43]. *Cyp2b9* and *Fmo3* both have strongly male-biased H3K27me3 marks across their gene bodies, which are lost in Ezh1/Ezh2-deficient liver (Fig 4A). Nevertheless, only *Cyp2b9* is depressed in the absence of Ezh1/Ezh2 (Fig 3A). The distinct responses of these two genes to Ezh1/Ezh2 loss raised the possibility that Ezh1/Ezh2--catalyzed deposition of H3K27me3 marks serves as the underlying mechanism for the repression of Class II but not Class I female-biased genes in male liver. Refuting this proposal, we found that although *Fmo3* was not de-repressed in Ezh1/Ezh2-deficient male liver, many other Class I female-biased genes were de-repressed, and at a frequency that matches their overall representation in the full set of female-biased genes (Fig 3B, *top* vs. *bottom*).

To better understand the GH-dependence of E1/E2-KO-responsive female-biased genes, we examined a set 113 robust female-biased genes (S2B Table), of which 65 (58%) are up-

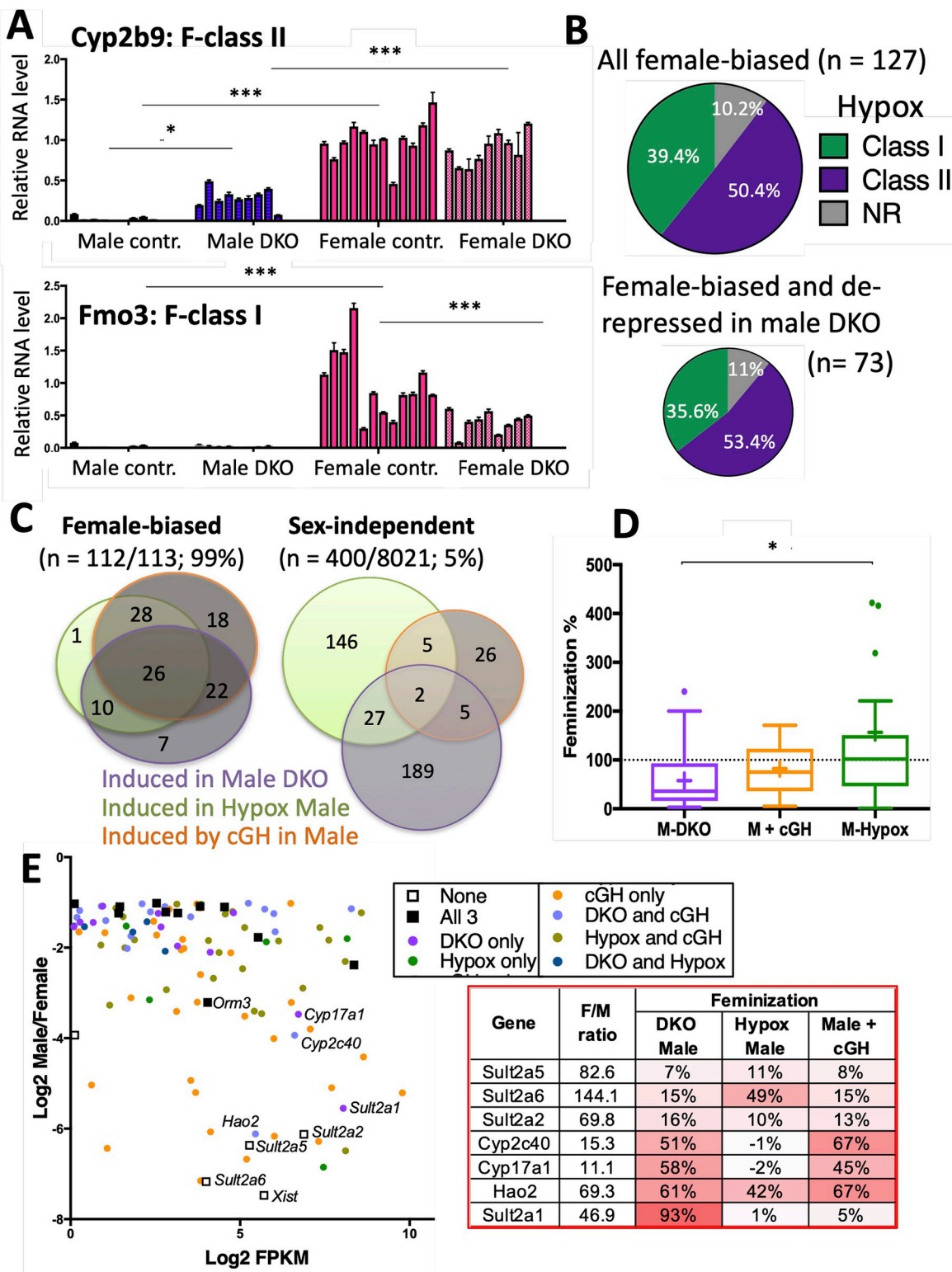

**Fig 3. Loss of Ezh1 and Ezh2 partially feminizes the expression of GH-responsive genes.** (**A**) Expression of *Fmo3*, a female-biased class-I hypophysectomy (Hypox)-responsive gene (i.e., gene is down-regulated in female liver after Hypox), and *Cyp2b9*, a female-biased class-II Hypox-responsive gene (i.e., gene is up-regulated in male liver after Hypox), in total RNA isolated from floxed control and E1/ E2-KO male and female mouse liver. Data shown are mean +/- SD values determined by RT-qPCR for n = 9–12 individual livers per group. The mean expression value for the control (wild-type) female group was set to 1. Significance values by ANOVA are shown: *, p<

0.05; ***, p< 0.001. Primers used for RT-qPCR analysis are shown in S1A Table. (**B**) Proportion of all female-biased genes (n = 127; female/male expression ratio > 2-fold) and proportion of the subset comprised of 73 female-biased genes that are up-regulated in E1/E2-KO male liver and that respond to Hypox and are either class I or class II female-biased genes, or that do not respond to Hypox (NR, not responsive). (**C**) Overlap analysis (*left*) for the set of female-biased genes (n = 113, male/female expression ratio > 2-fold, EdgeR FDR< 0.01, and FPKM >1 for control female vs control male mouse livers from the three models shown; see S2B Table) that are induced, either in E1/E2-KO male liver, following Hypox, or after continuous GH infusion for 14 days (cGH). Corresponding overlaps are also shown (*right*) for the set of stringently sex-independent genes, only 5% of which respond in one of the three mouse models, and where a much greater fraction of the genes that respond to E1/E2-KO do *not* also respond to Hypox or cGH treatment. (**D**) Boxplots showing the distribution of feminization values for 26 female-biased genes induced in male liver in all three mouse models (see panel C, center). Median value, horizontal line in each box; mean value, + sign within or above each box. Statistical significance by ANOVA: *, p < 0.05. (**E**) Graph, in the form of an MA plot, showing log2 (male/female ratio) vs. gene expression level, in log2 FPKM units, for the above set of 113 female-biased genes. Genes are colored, to indicate which mouse models/mouse treatments result in feminized expression of the gene in male liver by > 50%. Table at the right shows sex differences and percent feminization values for three *Sult2a* family genes that are largely resistant to feminization, and for four other highly female-biased genes that show substantial feminization in E1/E2-KO male liver. The strong feminization of *Sult2a6* and *Hao2* in hypophysectomized (Hypox) male liver indicates they are class-II female-biased genes. S2B Table shows the percentage feminization values for all 113 female-biased genes. DKO, Ezh1/Ezh2 double knockout mouse liver.

regulated in E1/E2-KO male liver. We assessed the responses of these 65 genes to two treatments that disrupt normal circulating GH patterns, hypophysectomy and continuous GH infusion. A large majority of the 65 genes (58 genes, 89%) were up-regulated either in livers of hypophysectomized male mice or in livers of male mice infused with GH continuously (Fig 3C). This can be compared to only 16% (34/211) of stringent sex-independent genes induced in E1/E2-KO male liver showing these responses. Moreover, 5 of the 7 female-biased genes that were not responsive to hypophysectomy or continuous GH infusion in male liver were actually repressed in livers of hypophysectomized female liver. Thus, almost all of the E1/E2-KO responsive female-biased genes (63 of 65; 97%) are apparently regulated by pituitary GH. Further, all but one of the 113 robust female-biased genes (*Xist)* was induced (i.e., feminized) in male liver in one or more of the three models examined (hypophysectomy, continuous GH infusion, and E1/E2-KO) (Fig 3C, *left*). However, the extent of feminization was substantially lower upon loss of Ezh1/Ezh2 (median feminization, 36%) than following hypophysectomy (median feminization, 102%) (Fig 3D; S2B Table). Moderately female-biased genes were often substantially feminized upon loss of Ezh1/Ezh2 (>50% feminization), whereas the mean feminization was only 29% for highly female-biased genes (F/M > 10). Exceptions include *Cyp2c40*, *Cyp17a1*, *Hao2* and *Sult2a1* (Fig 3E). Three highly female-biased sulfotransferases, *Sult2a2*, *Sult2a5* and *Sult2a6*, showed only partial feminization in all three models (Fig 3E), despite their substantial loss of Ezh1/Ezh2-dependent H3K27-trimethylation (Fig 4A, S4 Fig). Thus, for a subset of female-biased genes, the loss of Ezh1/Ezh2 and its capacity to form H3K27me3 repressive marks in male liver is not sufficient to de-repress gene expression. Repression of these female-biased genes in male liver likely involves mechanisms more complex than simply packaging the gene and its enhancers in H3K27me3 repressive chromatin.

## Loss of H3K27me3 marks at E1/E2-KO up-regulated female-biased genes

Global levels of H3K27me3 are reduced in 96% of E1/E2-KO hepatocytes at 3 months of age, without effects on non-parenchymal cells [33]. Here, we investigated the relationship between loss of H3K27me3 marks and the above changes in gene expression. First, to establish the validity of our sequencing results in the absence of a reference epigenome, we identified genomic regions where H3K27me3 marks were significantly lost (H3K27me3 sites responsive to E1/E2-KO), as well as regions where the intensity of H3K27me3 marks was unchanged in E1/E2-KO liver compared to control liver (static H3K27me3 sites; see Methods). qPCR analysis of the DNA obtained by chromatin immunoprecipitation (ChIP) confirmed that significant

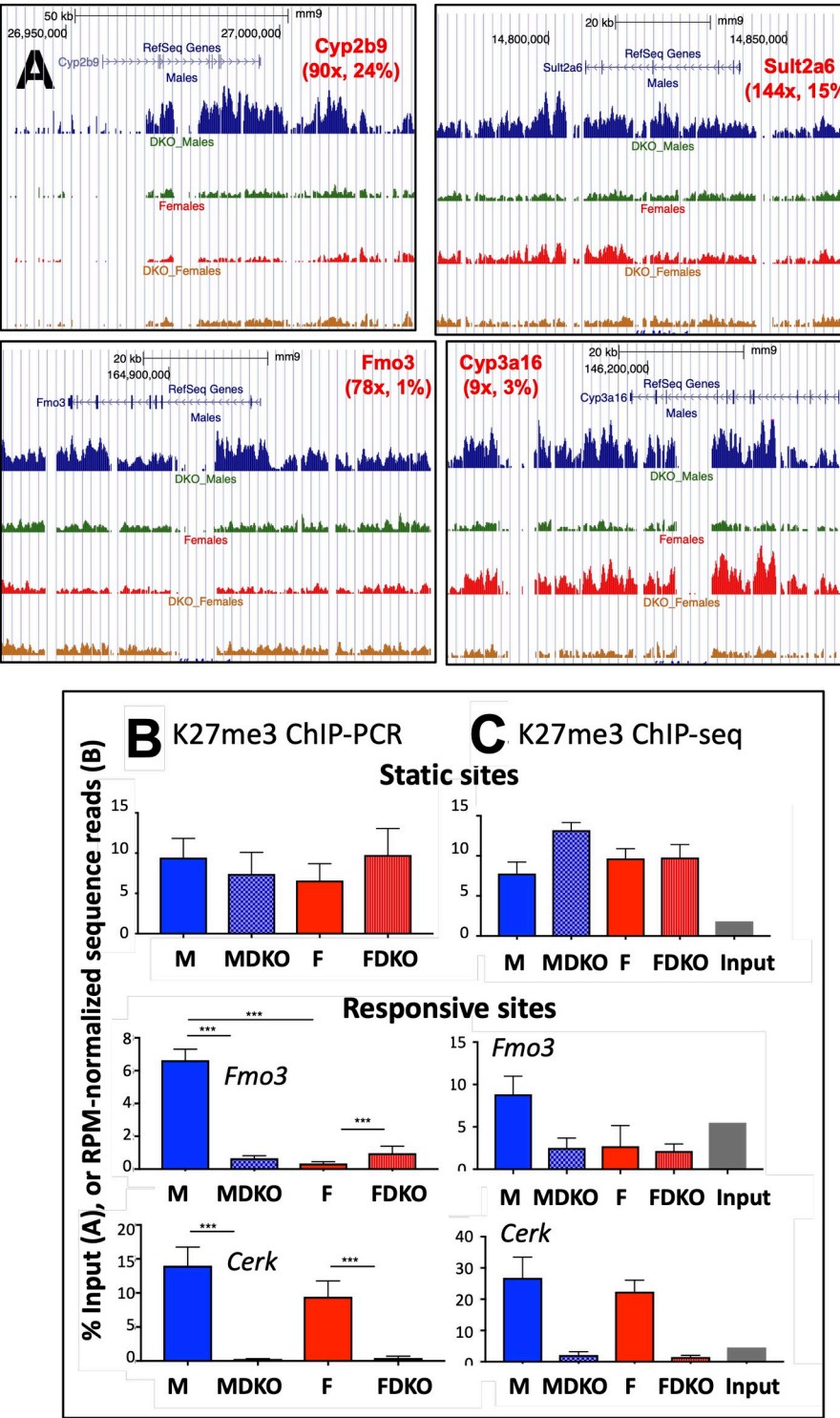

**Fig 4. Normalization of H3K27me3 ChIP-seq datasets.** (**A**) UCSC Browser screenshots showing loss of H3K27me3 sequence reads across the gene bodies of four female-biased genes. The male-bias in H3K27me3 read density in control liver (first vs. third track) but not in E1/E2-KO liver (DKO; second vs. fourth track) is also apparent. Values shown in parenthesis are: female to male expression ratio in wild-type mouse liver, and the percent feminization of gene expression in male E1/E2-KO liver (see data in S3 Fig). (**B**) ChIP-qPCR validation, and (**C**) reads per million (RPM) normalized read counts at a K27me3 static site (top row) and at two E1/E2-KO-responsive K27me3 differential sites identified by DiffReps (next two rows). The genomic regions interrogated map to a static intergenic region (top row;

qPCR amplicon: Chr 7, 53,631,603–53,631,654), to *Fmo3* (Chr15, 72,993,261–72,993,553) and to *Cerk* (Chr1, 164,912,550–164,912,800). The qPCR data shown correspond to % input values for n = 4 individual ChIP DNA samples per group, mean +/- SEM, with significance values determined by ANOVA with Tukey's multiple comparisons test (***, p< 0.001). (**C**) Data are shown for the indicated four mouse groups and the input control, as described in Methods. Read counts were obtained for the genomic location corresponding to the qPCR amplicon plus 100 bp, which approximates the 200 bp average sequence library insert size. Primers used for qPCR are shown in S1B Table. DKO, Ezh1/Ezh2 double knockout mouse liver.

changes in H3K27me3 mark intensity occurred at the responsive sites but not at the static sites, consistent with the ChIP-seq data for the same sites (Fig 4B vs. Fig 4C). At some responsive sites, the loss of H3K27me3 marks in E1/E2-KO liver indicated by sequencing was less complete than was indicated by qPCR analysis of the same ChIP'd DNA samples. Thus, our ChIP-seq data may underestimate the loss of H3K27me3 at some sites. Nevertheless, we were able to identify several thousand genomic regions with a significant difference in H3K27me3 marks between E1/E2-KO and control liver (S6B Table, S6C Table).

We compared control male and female liver and identified 538 genomic regions with male-biased H3K27me3 marks vs. only 11 regions with female-biased H3K27me3 marks (Fig 5A, S6A Table). This is consistent with the strong male-bias in mouse liver H3K27me3 marks reported previously [17]. However, the vast majority (97.4%) of the 21,036 H3K27me3 regions discovered in male and female liver did not shown significant sex differences. Strikingly, loss of Ezh1/Ezh2 abolished the sex bias at all 549 sex-dependent H3K27me3 sites; and furthermore, a total of 63 other H3K27me3 sites acquired sex-bias in the E1/E2-KO livers (Fig 5A, *right*). H3K27me3 marks were decreased at a majority (76%) of the sites dysregulated in E1/E2-KO liver, consistent with the loss of Ezh1/Ezh2. Twice as many H3K27me3 sites were dysregulated in male than in female E1/E2-KO livers (Fig 5B), consistent with the greater number of genes dysregulated in male liver (Fig 2). Sites where H3K27me3 marks were down-regulated were enriched for female-biased genes when compared to a background set of all liver-expressed genes, independent of whether the down-regulation was observed in E1/E2-KO livers from males or from females (S7 Fig).

Unexpectedly, H3K27me3 marks were up-regulated in E1/E2-KO liver at ~24% of all H3K27me3 differential sites (Fig 5B). The extent of up regulation at these sites was similar to the extent of down regulation at other H3K27me3 differential sites (Fig 5C). H3K27me3 sites up-regulated only in E1/E2-KO-female liver were enriched for female-biased genes (ES = 2.8, p = 5.38E-03; S7C Fig). The gene targets of the up-regulated H3K27me3 sites (452 genes; gene mapping based on annotations output by diffReps; see Methods) include 64 liver-expressed genes responsive to E1/E2-KO, 25 of which were repressed in either male or female E1/E2-KO liver (false discovery rate (FDR) < 0.05) (S6B Table, S6C Table). The increase in H3K27me3 marks at these sites could reflect histone mark changes in non-parenchymal cells, where the *Ezh2* gene is intact and presumably still active.

## Ezh1/Ezh2 loss is associated with gain of activating marks

H3K27 can be modified by acetylation to form H3K27ac, an activating mark associated with active enhancers [44]. In the absence of Ezh1/Ezh2, H3K27ac marks can increase and thereby reverse PRC2-mediated gene silencing [24, 45]. ChIP-seq analysis revealed significant increases in H3K27ac and a second activating mark, H3K4me1, at up to ~1,800 sites in male and female E1/E2-KO livers compared to sex-matched control livers; decreases in these activating marks were seen at many fewer (~100–150) sites (Fig 6A; see S6E–S6L Table for histone mark data). Thus, loss of the capacity to repress chromatin via H3K27-trimethylation is associated with an increase in activating histone marks. Further, whereas male-biased activating

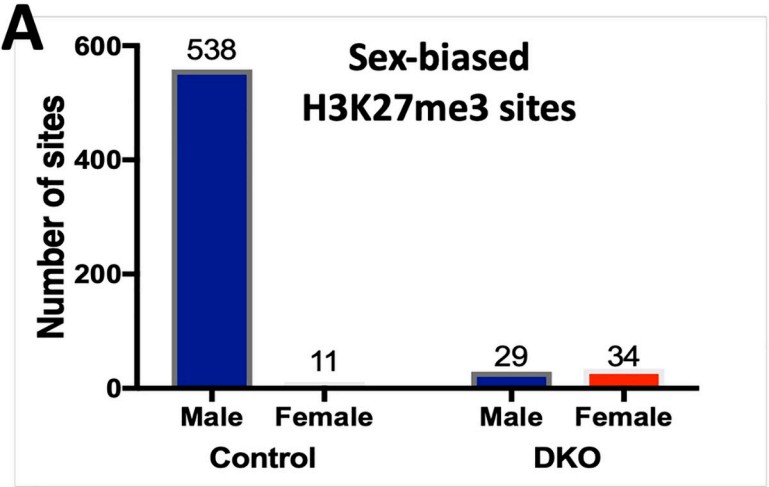

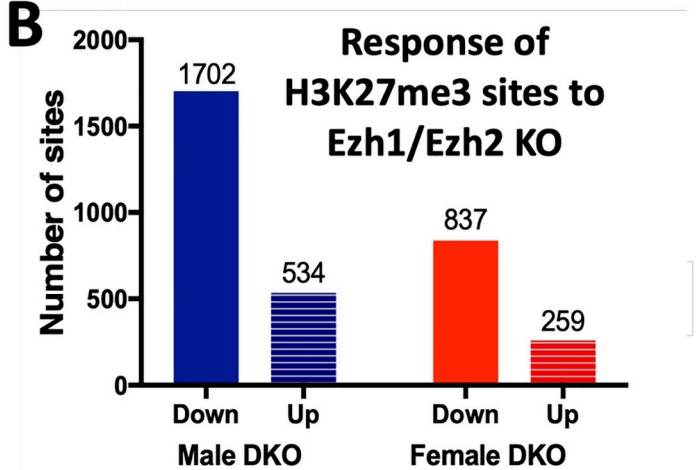

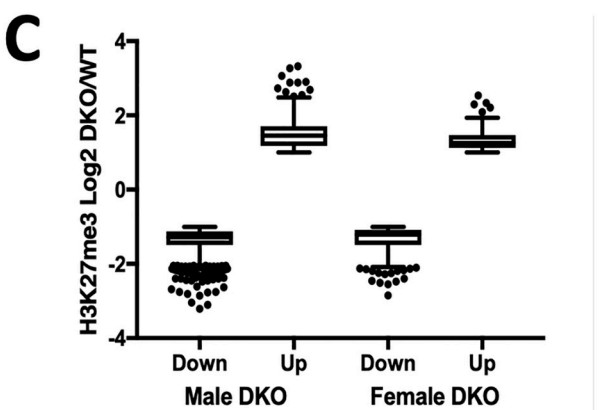

**Fig 5. H3K27me3 ChIP-seq. (A)** Number of sex-biased H3K27me3 sites found in control and in E1/E2-KO male and female mouse liver chromatin, as shown above each column. Sex-bias was lost at all 538 + 11 H3K27me3 sites in E1/E2-KO liver. 29 other H3K27me3 sites acquire male bias and 34 sites acquire female bias in the E1/E2-KO livers; the latter sets of sites are associated with 24 genes, of which only 5 are responsive to Ezh1/Ezh2 loss and only one is sex-biased. Also see S6A Table. **(B)** Number of H3K27me3 sites that were up-regulated or were down-regulated in E1/E2-KO male liver, or in E1/E2-KO female liver, when compared to control mice of the same sex, based on S6B Table and S6C Table. **(C)** Log2 of the fold-change for K27me3 signal for E1/E2-KO vs. control liver at those sites where the density of H3K27me3 marks decreases (Down) or increases (Up) in male and female E1/E2-KO liver. DKO, Ezh1/Ezh2 double knockout mouse liver.

chromatin marks (both H3K27ac and H3K4me1) were more than twice as frequent as female-biased activating marks in control livers, this sex difference was abolished in E1/E2-KO livers (Fig 6B). The overlap between the sets of sex-biased H3K27ac and H3K4me1 sites in control compared to E1/E2-KO mouse liver was low (S8 Fig), indicating that sex-biased chromatin marks are both gained and lost in Ezh1/Ezh2-deficient liver.

We mapped H3K27ac and H3K4me1 differential sites to their putative target genes using GREAT [46]. H3K27me3 differential sites were assigned to a gene if they overlapped the gene body or 3 kb surrounding its TSS. 846 genes up-regulated in E1/E2-KO males compared to control male liver were then classified into 8 groups based on the patterns of differential histone marks associated with each gene (S6M Table). Comparison of the distribution of histone mark patterns across the 8 groups for sex-biased genes to that of stringent sex-independent genes up-regulated in E1/E2-KO male livers (Fig 6C, *top* vs. *bottom*) revealed that genes that have differential H3K27me3 marks were more frequent in the sex-biased gene set (groups 2–4), while genes having differential H3K4me1 marks but not differential K27me3 marks were more frequent in the sex-independent gene set (groups 6 and 8). A majority (61–62%) of the genes up-regulated in E1/E2-KO male liver were not associated with any differential histone marks (group 1; Fig 6C). The up regulation of these genes in the absence of changes in H3K27me3, H3K27ac or H3K4me1 marks could be due to de-repression of their transcriptional activators. For example, 13% of the sex-biased genes in group 1 (no differential chromatin marks) are direct targets of the female-specific activator of female-biased genes CUX2 (S6M Table, column N), whose expression increases in E1/E2-KO male liver in association with loss of H3K27me3 and an increase in H3K27ac and H3K4me1 marks (S6M Table). Genes associated with increases in H3K27ac marks, either with or without induction of K4me1 marks (groups 6 and 7), showed significantly greater up regulation than genes without any differential marks (group 1) (Fig 6D, top). Female-biased genes with a loss of K27me3, either with or without induction of K4me1 (groups 3 and 5), showed greater increases in expression than genes having no associated differential marks (group 1).

Female-biased genes de-repressed in male E1/E2-KO liver that showed both a loss of H3K27me3 and a gain of H3K27ac marks had a significantly higher sex-bias than genes in other differential mark groups (Fig 6E, top, group 4 vs. groups 1,2,6,7,8). Highly female-biased genes (female/male expression ratio > 10) in group 1 (no differential marks) includes *Sult2a4*, which although it is induced by > 60-fold in E1/E2-KO male liver, only reaches 17% of the expression level of control female liver. However, this group also includes *Cyp2c40* and *Cyp17a1*, whose expression was significantly feminized in E1/E2-KO male liver (51% and 58%, respectively) (Fig 3E). Genes showing a gain in H3K27ac marks alone showed greater feminization than genes having no differential marks or loss of K27me3 alone (Fig 6E, bottom; group 7 vs. groups 1,3,4). Finally, 10 of 12 female-biased genes showing a loss of H3K27me3 marks and an increase in both H3K27ac and H3K4me1 marks (group 2) were fully feminized in male E1/E2-KO liver.

## Ezh1/Ezh2-dependent, sex-differential regulation of liver fibrosis genes and HCC-related genes

Male mice and humans are more susceptible to liver fibrosis than females [4–8] and show a male predominant incidence and progression of HCC [1, 2, 47]. Loss of liver Ezh1 and Ezh2 can impact the onset and progression of liver fibrosis, insofar as male E1/E2-KO livers acquire a nodular appearance with portal and periportal inflammation and collagen deposition by 8 months of age, along with substantial impairment of liver function [33]. Furthermore, we observed sex differences in liver pathology in E1/E2-KO mouse liver, with greater liver fibrosis,

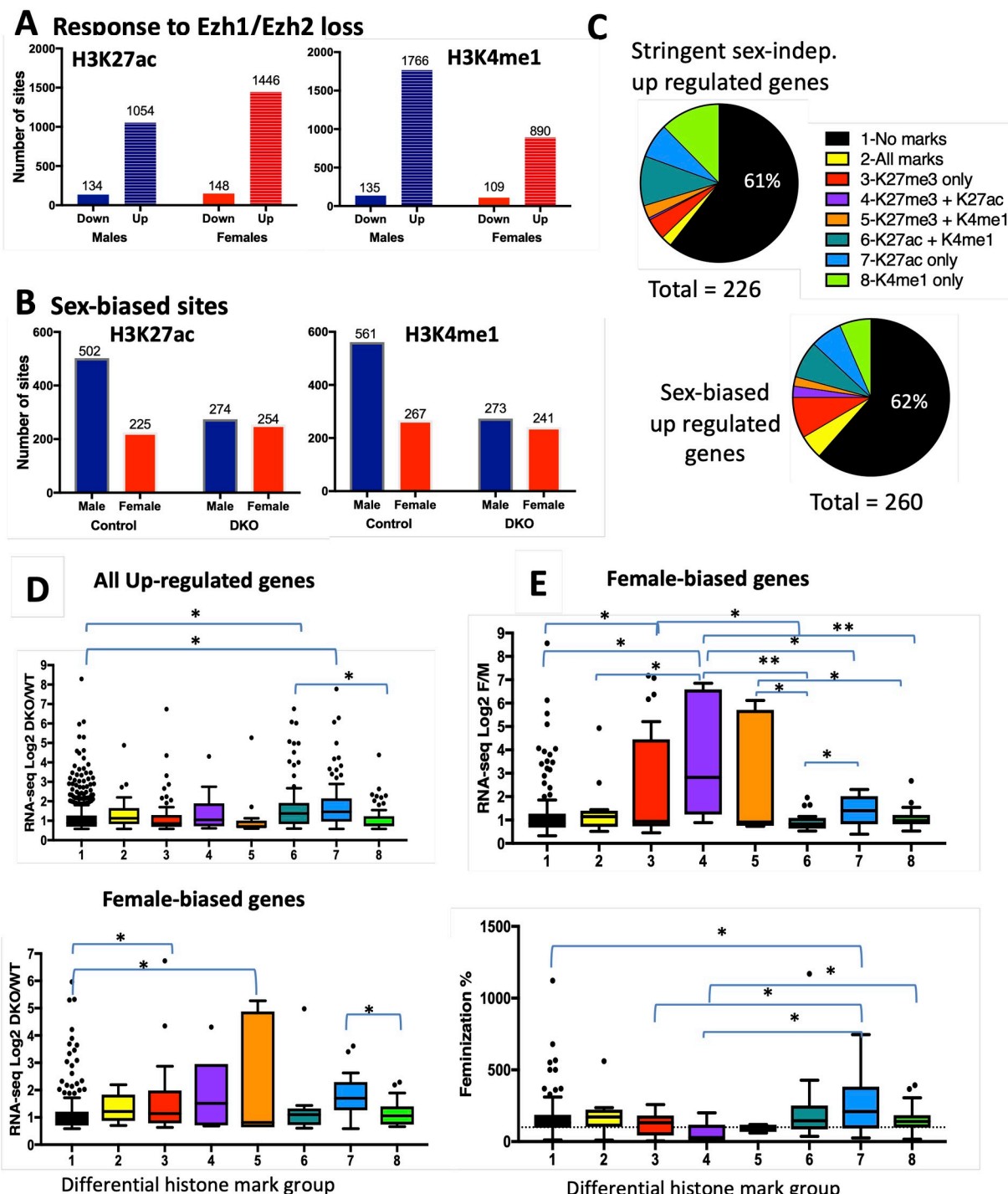

**Fig 6. De-repression of gene expression and responsive histone marks.** (**A**) Number of sites that show a significant decrease (Down) or a significant increase (Up) in each histone mark in E1/E2-KO male and E1/E2-KO female liver compared to sex-matched control liver. (**B**) Number of male-biased sites (Male) and number of female-biased sites (Female) for H3K27ac and H3K4me1 in control liver, and in E1/E2-KO liver. See listings in S6 Table. (**C**) Shown at the right are 8 different combinations of histone mark changes, one of which was assigned to each E1/E2-KO-responsive gene based on whether there was a decrease in H3K27me3 marks at either the gene body or promoter region, and/or an induction of H3K27ac marks or of H3K4me1 marks, as determined by GREAT analysis (see Methods). Shown is the histone mark pattern distribution for the 226 stringent-sex independent genes up-regulated in E1/E2-KO male liver (*upper pie chart*), or for the 260 sex-biased genes up-regulated in E1/E2-KO male liver (*lower pie chart*). These gene sets are based on data shown in Fig 2 and S3 Table for 846 genes up-regulated in male liver, of which 260 are sex-biased (154+86+16+4 = 260) and 226 are sex-independent (143 + 83 = 226). (**D**) *Upper* box plots show the distribution of log2 fold-change expression values for E1/E2-KO-male vs. wild-type male liver for the 846 genes up-regulated in male

E1/E2-KO liver in each of the 8 histone mark response groups shown in C. *Lower* box plots show the distribution of log2 fold-change expression values of E1/E2-KO-male vs. wild-type male liver only for the 244 female-biased genes up-regulated in male E1/E2-KO liver, for each of the 8 groups shown in C. (**E**) *Upper* box plots show the distribution of log2 male vs. female expression ratio for the 244 female-biased genes up-regulated in male E1/E2-KO liver, for each of the 8 groups shown in C. *Lower* box plots show the distribution of feminization percentages for the 244 female-biased genes up-regulated in the male E1/E2-KO livers, for each of the 8 groups shown in C. Significance values by t-test are indicated in each figure as follows: \*, p < 0.05; \*\*, p < 0.01; DKO, Ezh1/Ezh2 double knockout mouse liver.

nodular appearance and inflammatory cell infiltration seen in livers of male compared to female E1/E2-KO mice, and with female livers showing macro and micro-vesicular steatosis. In both sexes, the lobular structure was distorted (Fig 7A).

E1/E2-KO male livers also show increased susceptibility to the fibrogenic and hepatotoxic effects of carbon tetrachloride [33]. Strikingly, almost half of all liver fibrosis-associated genes (104 of 217 genes, 48%) and HCC-related genes (425 of 920 genes, 46%) were significantly changed in expression (FDR < 0.05, |fold-change| >2) in male E1/E2-KO compared to control liver at either 7 weeks or 8 months of age, or in livers of E1/E2-KO or control livers of male mice exposed to a regimen of carbon tetrachloride that induces hepatotoxicity and liver fibrosis (Fig 7B; S7 Table).

Similar numbers of fibrosis and HCC-related genes were induced in E1/E2-KO male livers at age 7 weeks as at 8 months of age (Table 1), even though no overt liver histopathological changes were apparent at 3 months of age [33]. A majority of the dysregulated genes were up-regulated (Table 1) and there was a significant increase in the degree of induction with advancing age (Fig 7C). Livers of E1/E2-KO male mice exposed to carbon tetrachloride showed the greatest number of responsive fibrosis and HCC-related genes, and the highest degree of up regulation (Table 1, Fig 7C), consistent with earlier findings based on a much smaller number of genes [33]. Moreover, the dysregulated sets of 104 liver fibrosis-related genes and 425 HCC-related genes were significantly enriched for female-biased genes and depleted of stringent sex-independent genes when compared to all liver-expressed genes (Fig 7D). More fibrosis and HCC-related genes were dysregulated in E1/E2-KO male compared to E1/E2-KO female liver (Table 1), consistent with the male bias in liver disease susceptibility. Similar patterns were seen for a set of 121 mouse orthologs of human genes that are up-regulated in advanced liver fibrosis [48] and for a set of 89 mouse orthologs of liver inflammation genes altered in human liver cirrhosis, a subset of which are associated with differences in HCC survival [49] (Table 1, S8 Table). We also identified 32 sex-independent genes that are more highly up-regulated in E1/E2-KO female than E1/E2-KO male mouse liver, and thus acquire female-biased expression in the absence of Ezh1/Ezh2 (Fig 2D, S9 Table). Three genes that show the most significant up regulation are *Igf2*, the lncRNA gene *H19*, and the microRNA *miR675* (Fig 7E). *Igf2* and *H19* are imprinted genes associated with HCC development [29, 50]. The *Igf2-H19* locus has a female-biased DNase hypersensitive site, identified previously [51], and a female-biased H3K4me1 mark that increases in female E1/E2-KO liver (Fig 7F). Thus, while there is moderate loss of H3K27me3 marks in both sexes, female-biased increases in activating marks may explain the strong induction of these genes seen in female liver (Fig 7F).

## Premature maturation of sex-biased liver gene expression in Ezh1/Ezh2-deficient liver

The combined deficiency of Ezh1/Ezh2 in mouse liver is reported to induce premature differentiation of perinatal hepatocytes, with 2 week postnatal Ezh1/Ezh2-deficient mice showing early increases in expression (early maturation) of a subset of genes not normally induced until 2 months of age in wild-type mice, including genes that regulate liver fibrosis [48]. Our reanalysis of the raw RNA-seq data from that study revealed that sex-biased genes are

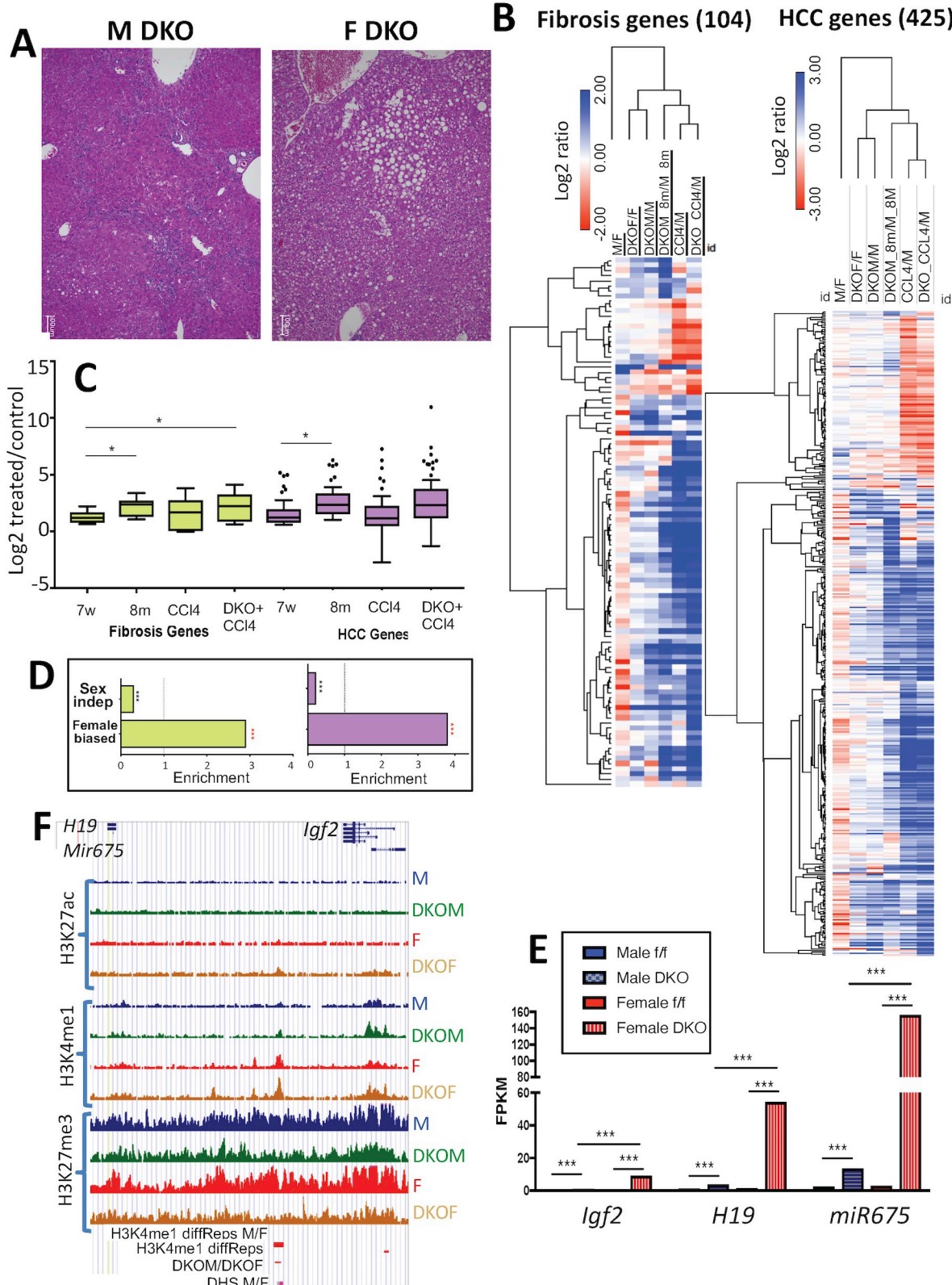

**Fig 7. Sex-biased dysregulation of liver fibrosis and HCC related genes. (A)** Hematoxylin and eosin staining of 8-month old male and female mouse E1/E2-KO liver sections, showing distortion of the lobular structure in both sexes. The male E1/E2-KO liver has a nodular appearance and inflammatory cell infiltration, while the female E1/E2-KO liver shows micro and macro vesicular steatosis. **(B)** Heat maps

showing log2 expression ratios for 104 E1/E2-KO responsive fibrosis-related genes and for 425 E1/E2-KO responsive HCC-related genes for the 6 indicated comparisons: wild-type male vs. wild-type female, E1/E2-KO female vs. wild-type female, E1/E2-KO male vs. wild-type male, 8-month-old E1/E2-KO male vs. 8-month-old wild-type males, males treated with carbon tetrachloride, and E1/E2-KO males treated with carbon tetrachloride compared to their age-matched controls. Color bars for ratios ranging ±2 or ±3 log2, as indicated. Average linkage hierarchical clustering implemented on the rows. (**C**) Box plots showing the distribution of log2 expression changes for 9 fibrosis-related genes (green) and 52 HCC-related genes (purple) that are up-regulated in both 7-week and 8-month E1/E2-KO male liver (S7 Table). Values shown for 7-week E1/E2-KO males, 8-month E1/E2-KO males, males treated with carbon tetrachloride and E1/E2-KO males treated with carbon tetrachloride compared to their age-matched controls, as described in Methods. Student's t-test: * $P < 0.05$; ** $P < 0.01$ and *** $P < 0.001$. (**D**) Enrichment, or depletion, of female-biased genes and stringent sex-independent genes for being in the set of dysregulated fibrosis-related or HCC related genes as compared to all liver expressed genes. (**E**) Expression of *Igf2*, *H19* and *miR675* determined by RNA-seq (FPKM values) in 7-week male and female control livers and in livers of male and female E1/E2-KO mice. Data shown are mean expression levels based on n = 4 individual livers per group. FDR values determined by EdgeR: ** $P<0.01$, and *** $P < 0.001$. (**F**) UCSC genome browser screenshot of the *Igf2-H19-Mir675* gene locus. Shown are normalized sequence read tracks for H3K27ac, H3K4me1 and H3K27me3 ChIP-seq data, for each of 4 groups, as marked on the right. This gene locus has a female-biased DNase hypersensitive site [51] flanked by a female-biased H3K4me1 site that is further induced in E1/E2-KO female liver (horizontal red bars at bottom), consistent with the greater gene induction seen in female liver. DKO, Ezh1/Ezh2 double knockout mouse liver.

significantly enriched in the set of all liver maturation genes (ES = 5.2, p<0.0001) (S10B Table), consistent with the widespread induction of sex-biased genes at puberty [52]. Moreover, we found that premature maturation occurred in both male and female Ezh1/Ezh2-deficient livers, but was significantly more extensive in males (240 genes prematurely up-regulated and 149 genes prematurely down-regulated in males, vs. 191 genes prematurely up-regulated and 90 genes prematurely down-regulated in females; p<0.0001, Fisher's exact test) (S10A Table). Importantly, sex-biased maturation genes were significantly enriched in the set of all premature maturation genes, with the strongest enrichments seen in Ezh1/Ezh2-deficient male liver (S10B Table). Finally, the number of genes showing sex-biased expression at 2 months of age decreased by >90% in this model of liver Ezh1/Ezh2-deficiency [48] (1022 vs. 69 sex-biased genes in 2-month wild type vs. E1/E2-KO liver at FDR <0.001; S10C Table), consistent with our findings, above, on the major role of Ezh1/Ezh2 in the regulation of sex bias gene expression in adult mouse liver.

## Discussion

Ezh1 and Ezh2 are epigenetic modifiers that catalyze H3K27-trimethylation essential for liver homeostasis and regeneration. Loss of Ezh1 and Ezh2 in hepatocytes leads to liver fibrosis, impaired liver function and increased susceptibility to the hepatotoxic effects of carbon tetrachloride [33]. Marked sex differences characterize the incidence, progression and severity of these liver pathologies, however, the underlying molecular basis for these sex differences in liver disease is only partially understood [53]. Our previous work identified H3K27me3 as a sex-biased epigenetic modification in mouse liver [17], suggesting that sex differences in Ezh1/

**Table 1. Number of fibrosis-related or HCC-related genes induced or repressed in E1/E2-KO livers compared to wild-type controls, and/or with each of the indicated CCl$_4$ (carbon tetrachloride) treatments.** Based on data in S7 Table and S8 Table.

| Differential gene expression comparison | Liver Fibrosis-related genes | | HCC-related genes | | Mouse orthologs of genes up-regulated in human liver fibrosis | | Mouse orthologs of inflammation genes dysregulated in human liver cirrhosis | |
|---|---|---|---|---|---|---|---|---|
| | # Induced | # Repressed | # Induced | # Repressed | # Induced | # Repressed | # Induced | # Repressed |
| E1/E2-KO Male vs. Male (7 wk) | 22 | 1 | 96 | 10 | 24 | 1 | 15 | 1 |
| E1/E2-KO Female vs. Female (7 wk) | 10 | 3 | 62 | 17 | 16 | 3 | 9 | 2 |
| E1/E2-KO Male vs. Male (8 mo) | 24 | 2 | 110 | 6 | 43 | 1 | 21 | 1 |
| CCl$_4$-treated Male vs. Male | 46 | 11 | 189 | 79 | 50 | 7 | 24 | 9 |
| E1/E2-KO Male + CCl$_4$ vs. Male | 63 | 10 | 245 | 68 | 65 | 3 | 45 | 7 |

Ezh2-catalyzed deposition of H3K27me3 marks functionally contribute to liver sex differences, including the striking differences in liver pathophysiology between the sexes. Here, we used male and female Ezh1/Ezh2 double knockout (E1/E2-KO) mice to address these questions by investigating the requirement of Ezh1 and Ezh2 for sex-biased gene regulation in mouse liver, and to discover any sex-dependent effects of Ezh1/Ezh2 loss on genes associated with liver disease. We found that hepatic Ezh1/Ezh2 deficiency induces a strong, preferential dysregulation of sex-biased genes, as compared to sex-independent genes. Many female-biased genes were significantly de-repressed in E1/E2-KO male liver in association with the loss of H3K27me3 marks across the gene body, while few male-biased genes were correspondingly de-repressed in E1/E2-KO female liver. In fact, many male-biased genes were down-regulated in E1/E2-KO male liver, which likely is a secondary response to the up regulation of female-biased genes expression. Ezh1/Ezh2-based repression of female-biased genes is thus a major epigenetic regulatory mechanism controlling sex-biased gene expression in male mouse liver. We also found that Ezh1/Ezh2 deficiency up regulates many genes associated with liver fibrosis and HCC in male liver, and that these changes are seen by 7 weeks of age, which precedes the histopathological changes seen in 8-month-old mice [33]. Finally, we found that liver fibrosis- and HCC-associated genes are differentially responsive to the loss of Ezh1/Ezh2 in male compared to female liver, which may contribute to the sex differences in disease incidence and progression.

Sex differences in liver gene expression are primarily regulated by sex-specific patterns of pituitary GH secretion. GH secretion is intermittent in males, whereas in females, pituitary GH release is more frequent, resulting in persistent stimulation of GH signaling in hepatocytes [9]. GH-responsive liver transcription factors, including STAT5b and two STAT5b-dependent repressors [13, 54], are key mediators of the sex-dependent transcriptional actions of GH, which operate in the context of GH-regulated sex differences in chromatin accessibility [51] and sex-biased chromatin states [17]. H3K27me3 is a strikingly sex-biased epigenetic regulatory modification that is specifically associated with strong repression of highly female-biased genes in male liver [17]. Consistently, female-biased genes were significantly enriched in the gene set up-regulated in E1/E2-KO male liver (Fig 2). However, the feminization of gene expression upon loss of Ezh1/Ezh2 was, in many cases, only partial. This contrasts with the more complete feminization achieved in two other mouse models that we examined, namely, continuous infusion of GH in male mice, which overrides the male, pulsatile plasma GH pattern and induces a majority of female-biased genes within 7 days [19], and ablation of pituitary hormone secretion by hypophysectomy, which de-represses Class II female-biased genes in male liver [43]. Thus, while Ezh1/Ezh2-catalyzed H3K27-trimethylation may repress female-biased genes in male liver, the loss of H3K27me3 marks alone is generally not sufficient for full gene activation, to normal female liver levels, and in some cases, is largely ineffective. For example, *Fmo3*, a highly female-specific gene, was not induced in E1/E2-deficient male liver, despite the extensive loss of male-biased H3K27me3 marks across its gene body. One possibility is that de-repression of *Fmo3* and other such genes requires both the loss of H3K27me3 marks and the acquisition of activating marks at distal enhancers, which may be subject to distinct epigenetic regulatory mechanisms [17]. Some sex-biased genes may be subject to more complex regulation, as exemplified by the female-biased gene, *Cyp17a1* (female/male expression ratio = 11-fold), which was substantially de-repressed in male E1/E2-KO liver (58% feminization) but showed an unexpected increase, rather than a decrease, in H3K27me3 marks in both male and female E1/E2-KO liver (S5 Fig).

Upstream Regulator analysis identified STAT5b as the most highly enriched negative regulator of the female-biased genes de-repressed in E1/E2-KO male liver, supporting the proposal that STAT5b regulates the deposition of H3K27me3 repressive marks at these genes in male liver, either directly or indirectly. Loss of H3K27me3 marks in E1/E2-KO male liver, and the

associated de-repression of female-biased genes, was often accompanied by increases in the active enhancer marks H3K27ac and H3K4me1. H3K27 acetylation, which is mutually exclusive with H3K27 trimethylation on a given nucleosome, is often enriched in the absence of PRC2 [55]; moreover, H3K27 acetylation prevents PRC2 binding and thereby antagonizes repression by PRC2 [24]. The increased expression of female-biased genes in E1/E2-KO male liver is therefore likely a direct result of de-repression caused by the loss of H3K27me3 marks combined with the subsequent gain in H3K27ac and other activating chromatin marks. Indeed, an increase in activating marks (H3K4me1 and H3K27ac) was associated with stronger induction of gene expression (Fig 6D, *top*, group 6 and group 7 vs group 1).

Our findings highlight the role of Ezh1/Ezh2-based repression of female-biased genes in male liver as an important mechanism to enforce liver sex differences. Overall, 37% of female-biased genes were significantly de-repressed in E1/E2-KO male liver (Fig 2C), indicating that Ezh1/Ezh2 is responsible–either directly or indirectly–for a substantial fraction of the epigenetic control of female-biased genes. Further, the actions of Ezh1/Ezh2 are sex-biased, with male liver showing many more genes dysregulated and more widespread loss of sites of H3K27-trimethylation than female liver. The absence of a sex bias in Ezh1/Ezh2 expression in adult liver (Fig 1), and our finding that ~97% of H3K27me3 peak regions do not show sex-differences in mark intensity, indicate that Ezh1/Ezh2 has similar intrinsic activity in livers of both sexes, and that other factors control the sex-specific actions of Ezh1/Ezh2 in adult mouse liver. Indeed, Ezh1/Ezh2-dependent gene repression is controlled by circulating GH patterns, whose continuous infusion in male mice induces loss of H3K27me3 marks at female-biased genes in association with their widespread de-repression in male liver [19]. Little is known about the molecular mechanisms by which the PRC2 complex and its Ezh1/Ezh2 catalytic subunit are recruited to their specific chromatin targets, in general, and specifically in this case, how plasma GH patterns regulate the sex-dependent interactions between PRC2 and its female-specific gene targets. PRC2 physically associates with several long non-coding RNAs (lncRNAs) [24], which may contribute to the target gene specificity of PRC2 action. Conceivably, some of the ~200 liver-expressed, nuclear lncRNAs that show sex-biased and GH-regulated expression in mouse liver [15] may contribute to the recruitment of PRC2 to female-biased genes repressed by Ezh1/Ezh2 in male liver. Gene co-expression network analysis of Diversity Outbred mouse liver has implicated eight sex-biased lncRNAs in negative regulation of genes of the opposite sex bias and inverse hypox-response class [56], which could potentially mediate some of the sex-biased effects of Ezh1/Ezh2 described here.

Several highly female-biased sulfotransferase genes, notably *Sult2a2*, *Sult2a5* and *Sult2a6*, failed to be substantially feminized upon loss of Ezh1/Ezh2 in male liver. Hepatic expression of these genes was also only partially feminized in male mice when the male, pulsatile pattern of GH stimulation was ablated by hypophysectomy, or when circulating GH profiles were feminized in continuous GH-infused male mice. The fact that feminization is only partial in the latter two mouse models, where GH signaling is disrupted, could be due to early postnatal effects of GH, which may irreversibly imprint (program) liver gene expression patterns [57]. In the case of *Sult2a5* and *Sult2a6*, although E1/E2-KO abolished the high levels of H3K27me3 marks seen across the gene body in wild-type male mouse liver (S4 Fig), gene expression in E1/E2-KO male liver only reached 7 to 16% of the normal, wild-type female level. Further study is needed to elucidate the mechanisms that establish any early, irreversible epigenetic differences in male liver, which may include GH-regulated DNA methylation of gene regulatory regions [58].

The present study used a double knockout mouse model, obtained by crossing *Ezh1* whole body knockout mice with *Ezh2*-hepatocyte specific knockout mice [33]. *Ezh2* null mutation results in lethality at early stages of mouse development [59], and while *Ezh1* null mutation

shows no decreased survival, only the combined loss of Ezh1 and Ezh2 in hepatocytes results in decreases in H3K27me3 [33]. E1/E2-KO male mice develop liver fibrosis at 8 months of age, and at 3 months of age, when histopathological abnormalities are not yet apparent, they show much greater susceptibility to the hepatotoxic effects of carbon tetrachloride as compared to control mice [33]. Here, we found that genes associated with liver fibrosis and HCC are significantly up-regulated in livers of male E1/E2-KO mice in an age-dependent manner, and following exposure to carbon tetrachloride (Fig 7), in a manner than correlates with the severity of the liver phenotype [33]. Male E1/E2-KO mouse liver also showed up regulation of mouse orthologs of human genes induced in advanced liver fibrosis [48] and of human genes related to liver inflammation whose expression is altered in human cirrhosis [49]. Genes up-regulated in common in advanced human liver fibrosis and in Ezh1/2-KO mouse liver include biomarkers of liver disease (e.g. Golm1) [60] and genes that play a fundamental role in liver fibrogenesis and HCC (e.g., Spp1/Osteopontin) [61]. E1/E2-KO female liver showed up regulation of the fewest number of fibrosis/HCC-related genes, consistent with the slower disease progression seen in female liver [62]. Moreover, H3K27me3 marks decreased and/or activating histone marks increased at 50% of the fibrosis- and HCC-related genes that were up-regulated in 7-week-old E1/E2-KO male liver, similar to the full set of E1/E2-KO up-regulated genes (S6M Table). Loss of H3K27me3 marks in human liver disease has been associated with increased transcription of genes associated with tumorigenesis [49], consistent with the association of Ezh1/2 loss with liver pathology seen in our Ezh1/2-KO mouse model.

Female-biased genes dominate the set of fibrosis/HCC-related genes that were up-regulated in E1/E2-KO male liver. This raises the question of why increased expression of these female-biased genes leads to an increase in liver fibrosis, in particular in male Ezh1/Ezh2-deficient mice [33]; whereas, in wild-type female liver, the higher expression of such genes compared to male liver is associated with decreased susceptibility to liver fibrosis and liver disease. The answer may relate to our findings on the effects Ezh1/Ezh2-deficiency on female-biased genes that can confer protection from HCC in wild-type female liver. One example is *Hao2*, which is down-regulated in HCC, and whose expression inversely correlates with metastasis and survival [63]. *Hao2* was significantly induced in E1/E2-KO male liver, but to only ~60% the level of control (wild-type) female liver, and this increase in expression may not be not sufficient to counteract the severe liver injury that results from the loss of Ezh1/Ezh2. Furthermore, other female-biased genes that have been identified as tumor suppressors in the context of HCC, such as *Trim24* [64, 65], were not up-regulated in male E1/E2-KO liver. Indeed, Upstream Regulator analysis showed that liver TRIM24 function was suppressed by Ezh1/Ezh2 deficiency, and that this suppression was more significantly associated with the set of all up-regulated genes in male than in female E1/E2-KO liver (p-value of overlap: 5.95E-20 vs. 7.74E-08) (S4C Table). Further studies are needed to determine the extent to which the higher expression of such liver disease-protective genes in female liver contributes to sex-differences in liver fibrosis and liver disease. Studies such as these take on added significance, given efforts to utilize Ezh2 inhibitors for treatment of HCC [66].

Several HCC-related genes that show sex-independent expression in wild-type liver were more strongly up-regulated by loss of Ezh1/Ezh2 in female than in male liver, i.e., they acquire female specificity in E1/E2-KO liver. These genes include *Igf2*, *H19* and *miR675* (Fig 7E, S11 Table). *Igf2* and *H19* are adjacent, imprinted genes [67] that show aberrant imprinting and epigenetic abnormalities in HCC [29]. *H19* is highly expressed in proliferative tissues, including liver regenerating after injury [67], and its first exon encodes miR-675, whose overexpression promotes liver cancer [50]. The strong up regulation of these three genes in E1/E2-KO female liver suggests female E1/E2-KO mice may show increased susceptibility to liver toxicity and

HCC compared to female wild-type mice, e.g., in response to hepatotoxins such as carbon tetrachloride.

In addition to the extensive up regulation of gene expression following loss of H3K27me3 marks, we identified a significant number of genes whose expression was down-regulated in the absence of Ezh1 and Ezh2, in both male and female liver. While H3K27me3 is generally regarded as a repressive mark, its deposition has also been associated with active transcription from promoters [68], whose activity loss could contribute to the gene down regulation we observed. In some cases, we saw an unanticipated increase in gene proximal H3K27me3 marks, which was associated with gene down-regulation (S7 Fig). Conceivably, the increase in H3K27me3 marks could be occurring in liver non-parenchymal cells (which do not express the *Alb-Cre* transgene needed for *Ezh2* deletion) as a secondary response to Ezh1/Ezh2 loss in hepatocytes. Finally, we observed an incomplete loss of H3K27me3 marks at many genomic locations. As PRC2 and its catalytic component, Ezh1/Ezh2, is the only known H3K27 methyltransferase in mammalian cells [69], this most likely reflects the continued expression of Ezh2 in liver non-parenchymal cells, and the persistence in hepatocytes of H3K27me3 marks deposited prior to the developmental induction of the *Alb-Cre* transgene [70].

We conclude that Ezh1/Ezh2-dependent H3K27me3 repressive marks play an essential, functional role in establishing and maintaining GH-regulated sex differences in liver gene expression. Loss of Ezh1 and Ezh2 in hepatocytes preferentially de-represses the expression of many female-biased genes in male mouse liver in association with the loss of H3K27me3 marks and the acquisition of activating histone marks. Significant differences in the regulation of liver fibrosis and HCC-related genes by Ezh1/Ezh2 are seen in male compared to female liver, which may contribute to the sex bias in liver disease progression.

## Methods

### Ethics statement

Livers used in this study were obtained from mice housed and handled according to the Animal Research Advisory Committee Guidelines of NIH (https://oacu.oir.nih.gov/animal-research-advisory-committee-guidelines), with all animal experiments approved by the NIDDK Animal Care and Use Committee (protocol ASP K089-LGP-17).

### Animal tissues

Livers from 7-week-old male and female *Ezh1*-knockout mice with a hepatocyte-specific knockout of *Ezh2* (E1/E2-KO mice, also designated Double-knockout (DKO) in the figures and tables) and their age and sex matched floxed littermate controls were generated as described [33]. Briefly, Ezh2$^{fl/fl}$ mice [71] were bred with *Alb-Cre* transgenic mice [70]; their offspring were bred with *Ezh1*-knockout mice (Dr. T. Jenuwein, Research Institute of Molecular Pathology, Vienna, Austria) [72] to generate E1/E2-KO mice. Livers from 2–8 week old male and female CD1 mice (ICR strain) were those described previously [52]. Hypophysectomy and continuous GH infusion of male mice for 14 d using an Alzet osmotic minipump were performed as described [19, 43].

### qPCR analysis

Liver total RNA (1 μg) was reverse transcribed using the Applied Biosystems High-Capacity cDNA Reverse Transcription Kit (Fisher, Cat#43-688-14). qPCR was performed using Power SYBR green PCR master mix and processed on an ABS 7900HT sequence detection system

(Applied Biosystems) or the CFX384 Touch Real-Time PCR detection system (Bio-Rad). For RT-qPCR, raw Ct values were analyzed using the comparative Ct method with normalization to the 18S RNA content of each sample. Primers used for qPCR are shown in S1A Table.

## RNA-seq analysis

Approximately 10% of each liver was snap frozen in liquid nitrogen and used to extract RNA with TRIzol reagent (Invitrogen Life Technologies Inc., Carlsbad, CA). Total liver RNA was isolated from each of 9 individual mouse livers per treatment group (Male (floxed) controls, Female (floxed) controls, Male E1/E2-KO and Female E1/E2-KO). Three RNA-seq libraries (biological replicates) were prepared for each treatment group; each sequencing library was comprised of a pool of RNAs obtained from n = 3 individual livers. Sequencing libraries were prepared using the Illumina TruSeq RNA library preparation kit (Illumina, cat# RS-122-2001) and 68 nt single-end sequence reads were obtained on an Illumina HiSeq instrument. RNA-seq data was analyzed using a custom pipeline [43]. Briefly, sequence reads were aligned to mouse genome build mm9 (NCBI 37) using Tophat (version 2.0.13) [73]. FeatureCounts [74] was used to count sequence reads mapping to the union of the exonic regions in all isoforms of a given gene (collapsed exon counting), and differential expression analysis was conducted using the Bioconductor package EdgeR [75]. We identified 11,491 liver-expressed genes, defined as genes expressed at >1 FPKM (Fragments Per Kilobase length of transcript per Million mapped reads) in at least one of the four sex–genotypes analyzed. 1,356 liver-expressed genes were significantly dysregulated in either male or female E1/E2-KO liver [i.e., EdgeR | fold-change| > 1.5 and adjusted p-value < 0.05 (i.e., 5% FDR, false discovery rate) for a comparison of E1/E2-KO male vs control male liver, or for a comparison of E1/E2-KO female vs control female liver]. A set of 1,131 genes showing significant male-female differences in expression in livers of floxed control mice in the E1/E2-KO background strain was identified using cutoff values for sex-differential expression of FDR < 0.01 and FPKM >1; these thresholds empirically corresponded to a >1.2-fold sex-difference in expression (S2A Table). Differential expression data from livers of male mice treated with GH given as a continuous infusion for 14 d, and for livers of hypophysectomized male and female mice, and their strain and age matched pituitary-intact control livers were obtained from S2 Table and S3 Table of [19] and from S3 Table of [43]. A set of 113 robust female-biased liver-expressed genes was defined as genes with a female/male expression ratio > 2-fold in control mice from each of three different mouse models (S2B Table). A set of 8,021 liver-expressed genes whose expression is stringently sex-independent was defined based on |fold-change| for sex-difference < 1.2 and FDR > 0.1 (S2C Table). The responses of these genes to the loss of Ezh1/Ezh2 are shown in S2A Table and S2C Table, and are summarized in S3 Table.

Raw RNA-seq data from GSE53627 [33] was analyzed using the custom pipeline cited above. Differential expression analysis was performed for the following comparisons: E1/E2-KO males vs. control males (8 months); males treated with carbon tetrachloride vs. control males; and E1/E2-KO-males treated with carbon tetrachloride vs. control males (S7 Table).

For female-biased genes, a percent feminization value was calculated based on each gene's response to each of the following treatments: Ezh1/Ezh2 loss in male liver; pituitary hormone ablation, as determined by hypophysectomy of male mice; and continuous infusion of male mice with GH for 14 days, as follows:

$$\% \text{ feminization} = \frac{100\% \left[\text{FPKM (treated male)} - \text{FPKM (control male)}\right]}{\left[\text{FPKM (control female)} - \text{FPKM (control male)}\right].}$$

Raw mouse liver RNA-seq reads from an independently developed mouse model of liver Ezh1/Ezh2 deficiency [48] were downloaded from GSE118757 (https://www.ncbi.nlm.nih.gov/gds/) and re-analyzed using our custom analysis pipeline. Differential expression analysis was performed for several comparisons, including: 2 month wild-type males + wild-type females (combined) vs. postnatal day 14 (P14) wild-type males + wild-type females (combined), to identify genes whose expression changes during liver maturation, as described in [48]; P14 Ezh1/Ezh2-deficient males vs. P14 wild-type males; and P14 Ezh1/Ezh2-deficient-females vs. wild-type females, at fold-change >2 and FDR <0.05 (S10A Table). Enrichment of sex-biased genes in the set of all liver maturation genes, and enrichment of sex-biased maturation genes that are prematurely expressed in P14-Ezh1/Ezh2-deficient male liver in the set of all prematurely expressed maturation genes was computed, as shown in S10B Table. Fisher's exact test was used to determine the statistical significance of all enrichments. Sex-bias in wild-type and in Ezh1/Ezh2-deficient livers at 2 weeks and 2 months of age was determined at FDR < 0.001 as shown in S10C Table.

## Chromatin preparation and chromatin immunoprecipitation (ChIP)

Chromatin was extracted from frozen liver tissue from each of 6 individual mice per group. Approximately 1 g of frozen liver was submerged in 4 ml of cross-linking buffer [10 mM HEPES (pH 7.6), 25 mM KCl, 0.34 M sucrose, 0.15 mM 2-mercaptoethanol, 2 mM $MgCl_2$, and Pierce protease inhibitor (1 tablet per 50 mL of buffer; ThermoFisher Scientific, cat. #A32965)] and homogenized using a glass dounce homogenizer. The homogenate was pushed thorough to a 70-micron cell strainer (Fisher Scientific. #22-363-548) using a 3 ml syringe plunger. The full volume (~ 5 ml) was transferred to a 15 ml conical tube containing 313 μl from a 16% formaldehyde ampule (ThermoFisher Scientific # 28906) mixed with 687 μl of crosslinking buffer, to give a final concentration of 0.83%. The samples were then incubated on a rocker for 5 min at room temperature. Cross-linking was halted by addition of 250 μl of a 2.5 M glycine (pH 8.0) solution (final concentration, 0.1 M), followed by incubation for 5 min at room temperature. Samples were pelleted (2,500 g for 5 min at 4˚C) and washed twice with 10 ml of PBS. Pellets were resuspended in 10 ml of Lysis Buffer 1 [50 mM HEPES (pH 7.5), 140 mM NaCl, 1 mM EDTA, 10% glycerol, 0.5% IGEPAL CA-630 (Sigma-Aldrich, cat. #I8896), 0.25% Triton X-100 (Sigma cat. #T8787) and Pierce protease inhibitor, as above], and incubated on a rocker for 10 min at 4˚C. Samples were centrifuged at 2,000 g at 4˚C for 5 min, the supernatant was removed, and the pellet was resuspended in 10 ml of Lysis buffer 2 [200 mM NaCl, 1 mM EDTA, 0.5 mM EGTA, 10 mM Tris-HCl (pH 8.0) and Pierce protease inhibitor] and rocked for 5 min at 4˚C. Samples were centrifuged at 2,000 g at 4˚C for 5 min, the supernatant was removed, and the pellet was resuspended in 2 ml of 1X radioimmunoprecipitation assay (RIPA) buffer [50 mM Tris-HCl (pH 8.1), 150 mM NaCl, 1% IGEPAL CA-630, 0.5% sodium deoxycholate] containing 0.5% SDS and Pierce protease inhibitor. Samples were sonicated for 20 cycles (30 s ON, 30 s OFF) using a Bioruptor Pico sonicator (Diagenode) in 15 ml Diagenode TPX tubes containing 0.3 ml polypropylene beads. A 15 μl aliquot of the sonicated chromatin was incubated at 65˚C for 6 hr to reverse cross-links. RNase A (ThermoFisher E053, 10 mg/mL) was added to a final concentration of 0.12 mg/mL and the samples then incubated at 37˚C for 30 min. Proteinase K (Bioline BIO-37084, 20 mg/mL) was added to a final concentration of 0.39 mg/mL and samples were digested at 37˚C for 2 h. A portion (8 μl) of each sample was analyzed by electrophoresis on a 1% agarose gel to size the DNA fragments, which mostly ranged from 100 to 300 bp. Reversed cross-linked DNA was quantified using a Quant-iT PicoGreen assay kit (Invitrogen). The remaining sonicated chromatin was snap-frozen in liquid nitrogen and stored at -80˚C until further use for ChIP. ChIP was performed as reported

previously [17, 18] using the following ChIP-validated antibodies: H3K27ac (Abcam cat. # ab4729, 3 μg antibody per 15 μg of sonicated chromatin), H3K4me1 (Abcam cat. # ab8895, 1.2 μg antibody per 15 μg of sonicated chromatin), H3K27me3 (Abcam cat. # ab6002, 2 μg antibody per 10 μg of sonicated chromatin), and normal rabbit IgG (Santa Cruz, cat. # sc-2027, 3 μg antibody per 15 μg of sonicated chromatin). ChIP DNA was quantified using a Quant-iT PicoGreen assay kit (Invitrogen) and analyzed by quantitative PCR (qPCR) using primers that interrogate genomic regions selected as positive controls or as negative controls for each of the histone marks based on our published ChIP-seq data [17].

## ChIP sequencing

Sequencing libraries were prepared for each of the above three histone marks using 20–50 ng of ChIP'd DNA. Libraries were prepared for each of 4 individual livers (biological replicates) for each sex and each genotype (male and female floxed control mice, and male and female E1/E2-KO mice; 16 libraries for each histone mark) using NEBNext Ultra II DNA Library Prep kit for Illumina (NEB, cat. # E7645). NEBNext Multiplex Oligos for Illumina (NEB, Set 1; cat. # E7335, NEB, Set 2; cat. # E7500) were used for multiplexing. The Agencourt AMPure XP system (Beckman Coulter, cat. # A63880) was used for sample and library purification. 50 nt paired-end sequence reads were obtained on an Illumina HiSeq instrument. ChIP-seq analysis was performed using a custom analysis pipeline initially developed for DNase-seq analysis and described elsewhere [76]. Individual biological replicates were validated using standard quality control metrics (FASTQC reports, confirmation of read length, and absence of read strand bias). FASTQ files for validated biological replicates were then concatenated to obtain a single set of combined reads for each condition. Sequence reads were mapped to the genome using Bowtie 2 (version 2.3.2) [77]. Genomic regions containing a significant number of H3K27me3 reads were identified using SICER (version 1.1, window size 400 bp and gap size 2400 bp) [78] and used for Reads in Peaks Per Million mapped sequence reads (RiPPM) normalization for UCSC browser visualization, as described below. A total of 21,036 H3K27me3 SICER peak regions were identified based on the union of peak regions discovered in male liver (16,862 regions) and female liver (19,795 regions). Genomic regions (peaks) enriched for H3K27ac and H3K4me1 sequence reads were discovered using MACS2 (version 2.1.1) [79] with default parameters. ChIP-seq peaks were visualized in the UCSC genome browser (https://genome.ucsc.edu/) after normalization of the genomic regions (i.e., ChIP-seq peak regions) discovered by SICER or MACS2 using RiPPM as a scaling factor, as follows. The peak lists identified for each sample (described above) were merged using mergeBed (BEDtools) to generate a single peak list (peak union). The fraction of reads in the peak union list for each sample was then calculated to obtain a scaling factor. Raw read counts were divided by the per-million scaling factor to obtain RiPPM normalized read counts.

## H3K27me3 differential peak discovery

DiffReps (version 1.55.4) [80] was used to identify genomic regions where H3K27me3 reads showed a significant difference in intensity between conditions being compared ('differential sites'). These analyses were based on pairwise comparisons of the 4 biological replicates per experimental group for each of the following comparisons: control males vs. control females; E1/E2-KO males vs. control males; E1/E2-KO females vs. control females; and E1/E2-KO males vs. E1/E2-KO females. diffReps differential windows were discovered using the nsd broad option of diffReps using each of four window sizes: 1 kb, 2 kb, 5 kb and 10 kb. For each analysis, the step size was set to 1/10 of the window size. Default statistical testing parameters of diffReps were used: negative binomial test with a p-value cutoff of < 0.0001 for significant

windows. Windows with significant differential H3K27me3 marks that were discovered with two or more of the window size settings were consolidated to eliminate redundancy by retaining the diffReps ID number and statistical information for the largest window size setting. Differential windows that were uniquely discovered by any of the four window size settings were also retained. The combined set of retained H3K27me3 differential windows were then filtered by the diffReps-determined values of FDR <0.05 and |fold-change| > 2 for the experiments groups being compared. The final lists of H3K27me3 differential sites are shown in S6A–S6D Table.

### H3K27ac and H3K4me1 differential peak discovery

DiffReps (see above) was then applied using the n = 4 biological replicate ChIP-seq samples for each chromatin mark, using the 1 kb window setting to identify diffReps differential sites. The differential sites identified were then filtered to retain those sites that overlap a MACS2-identified ChIP-seq peak. The resulting list of retained differential sites was further filtered for downstream analyses by excluding those sites that did not meet the threshold values of diffReps-determined FDR <0.05 and |fold-change| > 2 for the experimental groups being compared. The final lists of H3K27ac differential sites are shown in S6E–S6H Table, and the final lists of H3K4me1 differential sites are shown in S6I–S6L Table.

### H3K27me3 peak normalization

Due to the semi-quantitative nature of ChIP-seq methodologies [81], we first identified H3K27me3 regions that are largely unchanged across individuals and genotypes (static H3K27me3 sites). We used diffReps to identify stringent non-differential genomic windows (p-value > 0.1 and |fold-change| < 1.2) in the comparison of control male and E1/E2-KO male livers, and in the comparison of control female and E1/E2-KO female livers. The non-differential H3K27me3 sites identified in male liver were intersected with those identified in female liver, and 1,433 sites with 80% or greater reciprocal overlap across their lengths were retained. Those sites were then filtered by their average raw sequence read counts, and the top 35% of sites (502 sites, >400 average raw sequence reads per site) were retained. Those 502 sites were further filtered to retain the top 75% sites whose H3K27me3 marks were least variant across samples after RPM normalization (376 sites). ChIP-qPCR analysis of a subset of these stringent non-differential H3K27me3 sites (see Fig 4, below; primers used for qPCR are shown in S1B Table) confirmed that this approach does indeed identify invariant sites. ChIP-qPCR also revealed moderate compression of the strong differential sites in the ChIP-seq data, as expected. Next, we calculated the fraction of total sequence reads found in these 376 sites for each sample to obtain a scaling factor. Raw sequence read counts were divided by the per-million scaling factor to obtain normalized read counts for each sample. The normalization factor was then used to provisionally override the diffReps normalization results, and thereby obtain a new set of differential sites for each comparison. We observed high overlap (~93%) between the differential sites identified using the standard diffReps parameters, described above, and those identified when using the stringent non-differential site peak-based normalization factors, described in this paragraph. This high overlap validated our decision to use the standard diffReps normalization method to identify H3K27me3 differential sites for all downstream analyses.

### Mapping chromatin marks to genes

We used the output of diffReps to annotate and assign each H3K27me3 differential site (see above) to one of the following categories and to the genes associated with them:

ProximalPromoter (site within 0.25 kb of a transcription start site (TSS)), Promoter1k (site within 1 kb of a TSS), Promoter3k (site within 3 kb of a TSS), Genebody (site overlaps the genomic region extending from a gene's promoter to 1 kb downstream of the gene's transcript end site (TES)), Genedesert (genomic regions that are depleted of genes and are at least 1 megabase long), Pericentromere (region between the boundary of a centromere and the closest gene, excluding the proximal 10 kb of the gene's regulatory region), Subtelomere (defined in a manner similar to pericentromere), and OtherIntergenic (any region that does not belong to any of the above categories) (S6A–S6D Table). H3K27me3 differential sites annotated as ProximalPromoter, Promoter1k, Promoter3k and Genebody, and the genes associated to them, were used for downstream analyses. H3K27ac and H3K4me1 differential sites were mapped to their putative gene targets by GREAT [46] using the following parameters: each RefSeq gene was assigned a basal regulatory domain extending from 5 kb upstream to 1 kb downstream of the TSS, and the regulatory domain was extended in both directions to the nearest gene's basal regulatory domain up to a maximum of 1,000 kb in one direction [46]. For the results presented in Fig 6, genes that were up-regulated in male E1/E2-KO livers (846 genes) were classified into 8 groups (S6M Table) based on the GREAT gene-mark associations and the overlap of the full H3K27me3 region with the gene body or with the 3 kb genomic region surrounding the gene's TSS. Of the 846 genes, 226 are stringent sex-independent and 260 are sex-biased (S6M Table). The group classifications for these subsets of genes are shown in Fig 6C.

### Upstream regulator, functional annotation and pathway analysis

Differential gene expression and Upstream Regulator analysis for various sets of differentially regulated genes (S4 Table) were implemented using the IPA software suite (https://digitalinsights.qiagen.com/products-overview/discovery-insights-portfolio/analysis-and-visualization/qiagen-ipa/) (QIAGEN Inc). Gene Functional Classification analysis of differentially expressed genes was implemented using DAVID v.6.8 (https://david.ncifcrf.gov/); results from these analyses are presented in S5 Table. Genes related to liver fibrosis and hepatocellular carcinoma were obtained by searching for the terms "liver fibrosis" and "Hepatocellular carcinoma" in the Diseases and Functions search field. Lists output by IPA were further filtered to exclude chemicals by retaining only terms with an associated Entrez gene ID for mouse. Lists of 217 fibrosis-related genes and 920 hepatocellular carcinoma-related genes used in our analysis are shown in S7 Table. Heat map generation and clustering were carried out using Morpheus (https://software.broadinstitute.org/morpheus/), with average linkage hierarchical clustering implemented on the rows.

### Statistical analysis

The enrichment for up regulation of sex-biased genes in E1/E2-KO liver as compared to non-sex-biased genes was calculated from the ratio (A/B) divided by (C/D), as shown in this example: A = 240 female-biased genes up-regulated in male E1/E2-KO liver, and B = 842 *minus* 240 = 602 non-female-biased genes up-regulated in male E1/E2-KO liver, where 842 = total number of liver-expressed genes up-regulated in male E1/E2-KO liver; and C = 404 female-biased genes not up-regulated in male E1/E2-KO liver, and D = 10,847 liver-expressed genes not up-regulated in male E1/E2-KO liver (11,491 total liver-expressed genes *minus* 644 total female-biased genes). In this case, (A/B) divided by (C/D) = 10.7-fold enrichment Fisher exact test was used to determine the statistical significance of all the enrichment and depletion calculations (S12 Table). Graphical and statistical analyses were performed using GraphPad Prism 7 software. qPCR data are expressed as mean values and either standard errors of the mean or

standard deviation for n = 3 to 12 individual mouse livers per group, as specified in each figure legend. Unpaired t-test or one-way analysis of variance (ANOVA) was used to compare groups to each other, as noted in the figure legends.

## Supporting information

**S1 Fig. Western blot analysis of Ezh2 protein in male and female mouse liver.**
(TIF)

**S2 Fig. Heat map for the set of 440 E1/E2-KO-responsive sex-biased genes.** Shown are log2 fold-change values for each of the four indicated comparisons. Decrease in color intensity for many genes (column 4 vs. column 1) indicates loss of sex bias. DKO, Ezh1/Ezh2 double knock-out mouse liver.
(TIF)

**S3 Fig. E1/E2-KO-responsive female-biased genes. A, top.** Listing of expression data for genes whose UCSC Browser screenshots are shown in this figure. Data shown are fold-change and adjusted p-value for each of the comparisons indicated at the top. For those genes that are de-repressed in E1/E2-KO male livers, a feminization percentage and the differential histone mark group is indicated. **B.** UCSC screenshots of normalized RNA-seq or ChIP-seq reads for *A1bg* and *Hao2*, two female-biased genes that are induced in the E1/E2-KO male liver.
(TIF)

**S4 Fig. UCSC screenshots of normalized RNA-seq or ChIP-seq reads for *Sult2a1*, *Sult2a5* and *Sult2a6*, three female-biased genes induced in the E1/E2-KO male liver.**
(TIF)

**S5 Fig. UCSC screenshots of normalized RNA-seq or ChIP-seq reads for *Cyp17a1*, *Cyp2c69* and *Cyp2b9*, three female-biased genes induced in the E1/E2-KO male liver.**
(TIF)

**S6 Fig. UCSC screenshots of normalized RNA-seq or ChIP-seq reads for *Cyp3a16 a* female-biased genes induced in the E1/E2-KO female liver and *Fmo3* a female-biased gene not-induced in the E1/E2-KO.**
(TIF)

**S7 Fig. Sites where H3K27me3 marks were down-regulated in male and/or female E1/E2-KO livers.** (A) H3K27me3 sites that are up-regulated and down-regulated in E1/E2-KO male and E1/E2-KO female liver compared to sex-matched control livers, based on diffReps FDR <0.05, FC >2. Venn diagrams show the overlap between sets of sites for the indicated male-female comparisons. (B) Shown on the *left* are heat maps for H3K27me3 sites that are significantly differential only in E1/E2-KO male vs. control male comparison (E1/E2-KO-M unique differential sites) (first column in heat map) and RNA-seq expression ratios for their associated genes (next three columns). Similarly: *Right* heat map, H3K27me3 sites that are differential only in E1/E2-KO female vs. control female comparison (E1/E2-KO-F unique) and their associated genes; and *Middle* heat map, H3K27me3 differential sites common to E1/E2-KO males and E1/E2-KO females and their associated genes. Gene associations for each H3K27me3 site were based on the diffReps tool's output for those sites located in the gene body or promoter region. Values in the first column of each heat map represent log2 fold-change of the ChIP-seq signal between E1/E2-KO and control liver, and the values for each of the remaining 3 columns represent the log2 fold-change of the gene expression values between the indicated conditions, determined by RNA-seq. Shown above each heat map is the number

and percentage of sites that were associated with genes. (C) Shown are the enrichment scores and their p-values (see S12 Table) for female-biased genes associated with down-regulated or up-regulated H3K27me3 sites.
(TIF)

**S8 Fig. Sex-biased H3K27ac and H3K4me1 sites presented in Fig 6B.** Venn diagrams show the low degree of overlap between the sex-biased sites identified in control livers and those identified in E1/E2-KO livers. This low overlap between the sets of sex-biased H3K27ac and H3K4me1 sites in control, compared to E1/E2-KO mouse liver, indicates that sex-biased chromatin marks are both gained and lost in Ezh1/Ezh2-deficient liver. DKO, Ezh1/Ezh2 double knockout mouse liver.
(TIF)

**S1 Table. A.** RT-qPCR primers, used for analysis of the indicated mouse mRNAs. **B.** qPCR primers for validation of H3K27me3 static and H3K27me3 differential sites. Coordinates shown are for mouse genome mm9.
(XLSX)

**S2 Table. A.** Shown are 1,131 liver-expressed genes (FPKM >1) that showed a significant sex-bias in expression (EdgeR adjusted p-value < 0.01) in control (floxed) mouse liver. Also shown are the differential expression values obtained by EdgeR analysis for the following pair-wise comparisons: Males/Females (columns D-H); E1/E2-KO-males/floxed males (columns I-N); E1/E2-KO-females /floxed females (columns O-T), and E1/E2-KO-males/E1/E2-KO-females (columns U-Z). **B.** Shown is the set of 113 robust female-biased genes (female/male | fold-change| > 2-fold in each of three independent RNA-seq datasets: (1) floxed male vs. floxed female liver (liver samples G97 M1-M3/G97 M4-M6); (2) sham-treated male vs. untreated female liver (liver samples G88 M10-M12/G85 M5-M6) (both sets from GEO accession #GSE98586); and (3) untreated male vs. untreated female liver. **C.** 8,021 liver-expressed genes that show stringently sex-independent expression (FPKM >1, EdgeR adjusted p-value (FDR) >0.1, |fold-change| < 1.2).
(XLSX)

**S3 Table. A.** Responses of all liver-expressed genes (11,491 genes; FPKM >1) to loss of Ezh1/Ezh2 in male and female E1/E2-KO liver. The impact of E1/E2-KO on gene expression was evaluated at |fold-change| > 1.5 and FDR < 0.05. DKO, double-KO. **B.** Responses of sex-biased genes to loss of Ezh1/Ezh2 in male and female E1/E2-KO liver. Analyses are based on the 1,131 sex-biased genes shown in S2A Table. The impact of E1/E2-KO on gene expression was evaluated at |fold-change| > 1.5 and FDR < 0.05. **C.** Responses of stringently sex-independent genes to loss of Ezh1/Ezh2 in male and female E1/E2-KO liver. Analyses are based on the set of 8,021 stringent sex-independent genes shown in S2C Table. The impact of E1/E2-KO on gene expression was evaluated at |fold-change| > 1.5 and FDR < 0.05.
(XLSX)

**S4 Table. Upstream Regulator analysis, implemented in IPA, was used to discover upstream regulators of the indicated sex-biased and sex-independent gene sets.** Shown are those Upstream Regulators with a p-value of overlap < E-03 and an |activation Z-score| >2.
(XLSX)

**S5 Table. DAVID analysis of gene sets represented in Fig 2A, including sex-biased and stringent sex-independent genes that are up or down-regulated in DKO livers.**
(XLSX)

**S6 Table. A.** Sex-biased H3K27me3 sites. Shown are differential H3K27me3 sites in autosomes identified by diffReps comparison of male controls to female controls based on adjusted p-value < 0.05 and |Fold-change >2|. **B.** Differential H3K27me3 sites in autosomes identified by diffReps comparing E1/E2-KO males to male controls. **C.** Differential H3K27me3 sites in autosomes identified by diffReps comparing E1/E2-KO females to female controls. **D.** Differential H3K27me3 sites in autosomes identified by diffReps comparing E1/E2-KO males to E1/E2-KO females. **E.** H3K27ac sites showing significant differences in control male liver compared to control female liver (sex-biased H3K27ac sites). **F.** H3K27ac sites showing significant differences in E2-KO male liver compared to male liver controls. **G.** H3K27ac sites showing significant differences in E2-KO female liver compared to female liver controls. **H**. H3K27ac sites showing significant differences in E1/E2-KO male liver compared E1/E2-KO female liver. **I.** 3K4me1 sites showing significant differences in control male liver compared to control female liver (sex-biased H3K27ac sites). **J.** H3K4me1 sites showing significant differences in E2-KO male liver compared to male liver controls. **K.** H3K4me1 sites showing significant differences in E2-KO female liver compared to female liver controls. **L.**H3K4me1 sites showing significant differences in E2-KO male liver compared to E2-KO male liver. **M.** Up-regulated genes associated with differential histone marks. H3K27ac and H3K4me1 differential sites were mapped to their putative target genes using GREAT (McLean et al., 2010). H3K27me3 differential sites were mapped to their putative target genes if the region overlapped the gene body or the 3 kb genomic region surrounding the TSS (see Methods). 846 Genes up-regulated in E1/E2-KO males compared to control male liver were classified into 8 different groups based on the patterns of differential histone marks associated with each gene.
(XLSX)

**S7 Table. A.** Expression data for a set of 217 genes involved in liver fibrosis. Genes were obtained from IPA by searching for the term "Liver fibrosis" in the Diseases and Functions search field. **B.** Expression data for a set of 920 genes associated with hepatocellular carcinoma (HCC). Genes were obtained from IPA by searching for the term "Hepatocellular carcinoma" in the Diseases and Functions search field.
(XLSX)

**S8 Table. A.** Expression data for a set of 121 mouse orthologs of genes upregulated in advanced human liver fibrosis compared to early fibrosis. Gene list is from S7 Table of Grindheim et al. **B.** Expression data for a set of 90 human inflammation genes that are deregulated in liver cirrhosis. Gene list is from S4 Table of Hlady et al.
(XLSX)

**S9 Table. Listing of 82 genes that gain sex-bias in E1/E2-KO mouse liver.**
(XLSX)

**S10 Table. A.** Maturation genes identified from the RNA-seq datasets from Grindheim et.al. (GSE118757, PMID: 30689973). Differential expression analysis of data from male and females combined (database Exp. #734) identified 3,364 genes whose expression changes significantly from postnatal day 14 (P14) to 2 months of age (either up-regulated or down-regulated from P14 to 2 months, at FDR <0.05 and Fold-change >2. **B.** Enrichment score (ES) calculations. In section (I), we determined the enrichment of the set of 1,131 sex-biased genes (FPKM >1, FDR <0.01) in the set of all 3,364 maturation genes (S10A Table) compared to a background set of 8,021 stringent sex independent genes (FPKM >1, FDR >0.01, Fold-change <1.2). In section (II), we determined the enrichment of the subset of 466 sex-biased maturation genes that are prematurely expressed in P14-Ezh1/Ezh2-KO male liver in the set of 952 prematurely expressed maturation genes, as compared to a background set of stringently sex-independent

maturation genes. Enrichments are calculated separately for prematurely up-regulated genes, and for prematurely down regulated genes (top vs. bottom parts of (II). Enrichments were also calculated separately for all sex-biased genes, for female-biased genes, and for male-biased genes (first, middle, and last segments of (II)). Section (III), enrichments were calculated as in section (II), but was based on expression data in P14-Ezh1/Ezh2-KO female liver. Significance was assessed by Fisher's exact test. **C.** Sex biased genes at P14 and 2M, Wild-type and Ezh1/2-KO livers. RNA-seq data was obtained and re-analyzed from Grindheim et.al. (GSE118757, PMID: 30689973).
(XLSX)

**S11 Table. Imprinted genes that gain sex-bias in the E1/E2-KO.** A list of imprinted genes in the mouse was obtained from geneimprint.com. Shown are 5 imprinted genes that gain sex-bias after E1/E2-KO. Only *H19* and *Igf2* become significantly female-biased due to their preferential up-regulation in female E1/E2-KO liver.
(XLSX)

**S12 Table. Contingency tables for the enrichments shown in Fig 2, Fig 5, Fig 7 and S7 Fig.**
(XLSX)

## Acknowledgments

We thank Dr. Andy Rampersaud and Kritika Karri of the Waxman Lab for developing and implementing the sequencing analysis pipeline used in this study.

## Author Contributions

**Conceptualization:** Dana Lau-Corona, David J. Waxman.

**Data curation:** Dana Lau-Corona.

**Formal analysis:** Dana Lau-Corona.

**Funding acquisition:** Lothar Hennighausen, David J. Waxman.

**Investigation:** Dana Lau-Corona, Woo Kyun Bae, David J. Waxman.

**Project administration:** David J. Waxman.

**Resources:** Woo Kyun Bae, Lothar Hennighausen.

**Supervision:** David J. Waxman.

**Visualization:** Dana Lau-Corona, David J. Waxman.

**Writing – original draft:** Dana Lau-Corona, David J. Waxman.

**Writing – review & editing:** Dana Lau-Corona, David J. Waxman.

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
