## [Decision Letter · Decision Letter 0]

23 Jan 2020

Dear Dr Waxman,

Thank you very much for submitting your Research Article entitled 'Sex-biased genetic programs in liver metabolism and liver fibrosis are controlled by EZH1 and EZH2' to PLOS Genetics. Your manuscript was fully evaluated at the editorial level and by independent peer reviewers. The reviewers appreciated the attention to an important problem, but raised some substantial concerns about the current manuscript. Based on the reviews, we will not be able to accept this version of the manuscript, but we would be willing to review again a much-revised version. We cannot, of course, promise publication at that time.

If you decide to revise the manuscript for further consideration at PLOS Genetics, please aim to resubmit within the next 60 days, unless it will take extra time to address the concerns of the reviewers, in which case we would appreciate an expected resubmission date by email to plosgenetics@plos.org.

[LINK]

We are sorry that we cannot be more positive about your manuscript at this stage. Please do not hesitate to contact us if you have any concerns or questions.

Yours sincerely,

Xiao-bo Zhong, Ph.D.

Guest Editor

PLOS Genetics

Wendy Bickmore

Section Editor: Epigenetics

PLOS Genetics

Reviewer's Responses to Questions

**Comments to the Authors:**

Reviewer #1: In this manuscript, the authors investigated the functional impact of sex differences in histone protein modifications related to sex-biased liver diseases. The work was done using mouse models deficient in two enzymes, Ezh1/Ezh2, which form the histone repressive mark H3K27me3. Through RNA-seq, ChIP, and qPCR analyses as well as bioinformatics approaches, the authors found that loss of hepatic Ezh1/Ezh2 disrupts sex-biased H3K27me3 repressive marks, and abolishes the control of sex-biased genes with increased expression in male mouse liver for many female-biased genes and decreased expression of male-biased genes, including those that are involved in liver fibrosis and hepatocellular carcinoma (HCC). This information is interesting and informative to the research community. The manuscript was well written. Most of the data are of high quality. The following concerns need to be addressed:

1. Based on the qPCR data, the expression of Ezh1 and Ezh2 between male and female were not much difference. If Ezh1/Ezh2 indeed control sex-biased difference in hepatic gene expression between male and female, please address why there is no difference in hepatic Ezh1 and Ezh2 expression between male and female? And also, expression levels of Ezh1 and Ezh2 proteins need to be examined, as mRNA and protein levels are not consistent in many cases.

2. There are significant differences in gene expression and pathogenesis in related to hepatic fibrosis and HCC between human and mouse. To translate the findings from the animals, validation of Ezh1/Ezh2-controlled, sex-biased H3K27me3 repressive marks with human liver samples or genomic data is required.

Reviewer #2: Note: This review is also uploaded as an attachment.

Summary

The sexual dimorphic hepatic gene expression profile is linked to the sex-biased incidence and severity of liver diseases, such as liver fibrosis and HCC. This manuscript probed an essential role of EZH1 and EZH2, the catalytic components of the polycomb repressive complex-2 (PRC2), for their regulation of the sex-based repressive chromatin histone mark H3K27me3 and the metabolic and fibrosis gene expressions profile in the liver.

This study established that Ezh1/2-based repression of female-biased genes is a major epigenetic mechanism that controls sex-biased gene expression in the male liver and represented a direct demonstration of the functional importance of the sex-biased repressive histone mark H3K27me3.

The author provided comprehensive characterization of the hepatic chromatin modification states such as H3K27me3, H3K27ac and H3K4me1, in combination with the previously published datasets from hypophysectomy and continuous GH infusion models. The up regulation of the genes linked to liver fibrosis and HCC is more significant in the male liver but to a lesser extent in the female liver, which may indicate the clinical relevance of Ezh1/2 in patients with liver disease, however, the functional evidence of this observation in vivo is weak.

The study design was clear and straight-forward and the manuscript was well polished, but there are some concerns for which additional experiments and revision of the manuscript should be considered.

Major points:

1. Since (i) deletion of Ezh2 in embryonic liver impairs hepatic progenitor expansion, thus the liver differentiation and maturation (ref 25), (ii) Alb-Cre transgene is also active in fetal and neonatal mice (Ref 63), (iii) a more recent study reported that disruption of Ezh1 and Ezh2 in livers caused perinatal hepatocytes to differentiate prematurely (Grindheim et al, Gastroenterology, 156:1834-1848, 2019), whether the combined hepatic Ezh1/Ezh2 deficiency model in this study have similar issue? If so, is this defect also sex-specific? And whether the sex-specific differences in the adult mice of current study are secondary to this developmental defect? And how is this defect linked to the context of sex-specific growth hormone regulation?

2. The author showed the Ezh1/2-dependent and sex-biased regulation of liver fibrosis and HCC-related gene expression in Fig. 7. However, this is largely based on the gene expression analysis, further functional evidence is necessary for the author to draw a solid conclusion. As previous reports mainly used the male mice in their studies (Ref 33 and Grindheim et al, Gastroenterology, 156:1834-1848, 2019), a comparative characterization of the liver fibrosis and damage phenotype between male and female would be informative at this point, either in the older hepatic Ezh1/Ezh2 deficiency mice or in the CCl4-induced hepatotoxity model.

Minor points:

3. Fig.2 and 3 are very busy - need to be reorganized or streamlined to make them easier to follow.

4. Fig.3 - To what extent does EZH1/2-dependnent liver chromatin repression contribute to the GH-based regulation on the sex-based gene expression profile? Please clarify in the text.

5. Regarding Discussion section (line 503) - the incomplete loss of H3K27me3 marks at many genomic loci, are there any other additional genes or enzymes that could catalyze the H3K27me3 mark formation? Since Alb-Cre transgene is also active in fetal and neonatal mice (Ref 63).

Reviewer #3: This interesting manuscript has utilized Ezh1/Ezh2 hepatocyte double null mice to demonstrate the necessity of H3K27me3 in controlling the sex-specific gene expression that are relevant to liver diseases (such as liver fibrosis and cancer) as well as following toxicant exposure. Comprehensive integrations of RNA-Seq and ChIP-Seq datasets were performed taking the advantage of data generated de novo (the DKO mice) as well as from online data repositories. The authors concluded that the Ezh1/Ezh2 catalyzed H3K27me3 regulates sex dependent genetic programs in liver metabolism and fibrosis through sex-dependent effects on the epigenome and may thereby determine the sex bias in liver disease susceptibility. The findings are novel and important in further understanding the mechanisms of gender specific disease phenotypes.

Major comments:

1) The mouse model used was an Ezh1 whole body knockout crossed with Ezh2 hepatocyte specific knockout mice. Please discuss why Ezh1 hepatocyte specific mouse was not used to cross with the Ezh2 hepatocyte specific knockout mice? Is it because it is embryonically lethal?

2) According to NCBI Gene RNA-Seq repository, Ezh1 expression in liver is 3.057 (FPKM), whereas Ezh2 expression in liver is only 0.272 (FPKM). Similar trend of gene expression profiles was also observed in the present study (Figure 1A and 1B). It appears that Ezh2 is barely expressed in liver. This is probably why in DKO there was no difference in the basal expression of Ezh2 as compared to controls. Could extrahepatic depletion of Ezh2 be more important in indirectly modulating the sex-specific gene expression in liver through altering remote hormonal signaling?

3) Figure 1A: it is not clear exactly how the data were expressed for RT-qPCR data. Were they normalized to housekeeping gene and expressed as % of housekeeping genes? Also it will be helpful to include the Cq values.

4) Figure 2. It is very interesting that hepatic Ezh1/Ezh2 deficiency down-regulates many male biased genes, and the authors suggested that this was due to a secondary response to the disruption of female-biased gene expression. However, it will be even more interesting to conduct mechanistic investigations regarding how this secondary response occurs. e.g. is there any enhanced DNA methylation at these gene loci? Also it will be interesting to conduct up-stream regulator analysis at these down-regulated genes and see what factors may serve as repressors of these genes in the absence of H3K27Me3 enzymes.

5) The authors focused on liver fibrosis and HCC pathways in Figure 7. However, in Figure 2, it would be interesting to know in general what pathways (e.g. KEGG) are affected using a top-down approach.

6) It is very interesting that Ezh1/Ezh2 deficiency increased Cyp2b9 mRNA in males, but did not increase Fmo3 mRNA in males; however, in Fmo3 gene loci, H3K27me3 was also markedly decreased in DKO males (similar to Cyp2b9). This indicates other suppressive marks are present – it would be interesting to explore this.

Minor comments:

1) I suggest changing “sex-biased genes” to “sex-enriched” or “sex-specific” genes. Biased usually refers to opinions.

2) CCl4 is not a hepatotoxin, it is a hepatotoxicant.

**Have all data underlying the figures and results presented in the manuscript been provided?**

Reviewer #1: Yes

Reviewer #2: Yes

Reviewer #3: None

PLOS authors have the option to publish the peer review history of their article (what does this mean?). If published, this will include your full peer review and any attached files.

Reviewer #1: No

Reviewer #2: No

Reviewer #3: No

---

## [Decision Letter · Decision Letter 1]

24 Apr 2020

Dear Dr Waxman,

We are pleased to inform you that your manuscript entitled "Sex-biased genetic programs in liver metabolism and liver fibrosis are controlled by EZH1 and EZH2" has been editorially accepted for publication in PLOS Genetics. Congratulations!

Yours sincerely,

Xiao-bo Zhong, Ph.D.

Guest Editor

PLOS Genetics

Wendy Bickmore

Section Editor: Epigenetics

PLOS Genetics

Comments from the reviewers (if applicable):

Reviewer's Responses to Questions

**Comments to the Authors:**

Reviewer #1: The previous comments were addressed.

Reviewer #2: The revised manuscript addressed my major concerns of the original submission.

In regard that “the sex-specific differences seen in the adult DKO mice of the current study are secondary to the developmental defect reported by Grindheim et al.”, an inducible hepatic EZH1/2 knockout model will be necessary to further address the sex-specific effect of EZH1/2 deficiency in adult or aged mice in future study.

**Have all data underlying the figures and results presented in the manuscript been provided?**

Reviewer #1: Yes

Reviewer #2: Yes

PLOS authors have the option to publish the peer review history of their article (what does this mean?). If published, this will include your full peer review and any attached files.

Reviewer #1: No

Reviewer #2: No

**Data Deposition**

http://datadryad.org/submit?journalID=pgenetics&manu=PGENETICS-D-19-01963R1

**Press Queries**

---

## [Editor Report · Acceptance letter]

12 May 2020

PGENETICS-D-19-01963R1 

Sex-biased genetic programs in liver metabolism and liver fibrosis are controlled by EZH1 and EZH2 

Dear Dr Waxman, 

We are pleased to inform you that your manuscript entitled "Sex-biased genetic programs in liver metabolism and liver fibrosis are controlled by EZH1 and EZH2" has been formally accepted for publication in PLOS Genetics! Your manuscript is now with our production department and you will be notified of the publication date in due course.

With kind regards,

Jason Norris

PLOS Genetics

On behalf of:
